# Modeling medulloblastoma in vivo and with human cerebellar organoids

Claudio Ballabio [1,9], Marica Anderle[1,9], Matteo Gianesello[1], Chiara Lago[1], Evelina Miele[2], Marina Cardano [3], Giuseppe Aiello[1], Silvano Piazza[3], Davide Caron[1], Francesca Gianno [4,5], Andrea Ciolfi[6], Lucia Pedace[2], Angela Mastronuzzi [2], Marco Tartaglia[6], Franco Locatelli[2,7], Elisabetta Ferretti [8], Felice Giangaspero[4,5] & Luca Tiberi[1]*

Medulloblastoma (MB) is the most common malignant brain tumor in children and among the subtypes, Group 3 MB has the worst outcome. Here, we perform an in vivo, patient-specific screen leading to the identification of *Otx2* and *c-MYC* as strong Group 3 MB inducers. We validated our findings in human cerebellar organoids where Otx2/c-MYC give rise to MB-like organoids harboring a DNA methylation signature that clusters with human Group 3 tumors. Furthermore, we show that SMARCA4 is able to reduce Otx2/c-MYC tumorigenic activity in vivo and in human cerebellar organoids while SMARCA4 T910M, a mutant form found in human MB patients, inhibits the wild-type protein function. Finally, treatment with Tazemetostat, a EZH2-specific inhibitor, reduces Otx2/c-MYC tumorigenesis in ex vivo culture and human cerebellar organoids. In conclusion, human cerebellar organoids can be efficiently used to understand the role of genes found altered in cancer patients and represent a reliable tool for developing personalized therapies.

---

[1] Armenise-Harvard Laboratory of Brain Cancer, Department CIBIO, University of Trento, Via Sommarive 9, 38123 Trento, Italy. [2] Department of Pediatric Hematology/Oncology and Cellular and Gene Therapy, Bambino Gesù Children's Hospital, IRCCS, Rome, Italy. [3] University of Trento, Via Sommarive 9, 38123 Trento, Italy. [4] Department of Radiologic, Oncologic and Anatomo Pathological Sciences, University Sapienza of Rome, Rome, Italy. [5] IRCCS Neuromed, Pozzilli, Isernia, Italy. [6] Genetics and Rare Diseases Research Division, Ospedale Pediatrico Bambino Gesù, IRCCS, 00146 Rome, Italy. [7] Department of Pediatrics, Sapienza, University of Rome, Rome, Italy. [8] Department of Experimental Medicine, Sapienza University, Rome, Italy. [9] These authors contributed equally: Claudio Ballabio, Marica Anderle. *email: luca.tiberi@unitn.it

Medulloblastoma (MB) is one of the most aggressive brain tumors affecting children and stands as a cause of a high percentage of morbidity and mortality among cancer patients[1]. MB is a biologically and clinically heterogeneous tumor, including several subgroups[2]. Among the groups, patients with Group 3 MB (characterized by *c-MYC* upregulation), have the worst outcome with ~50% of the tumors metastatic at the time of diagnosis. The currently available therapy for MB consists of maximal safe resection, craniospinal radiation (for children ≥ 3 years old) and chemotherapy. Therefore, developing humanized mouse model of Group 3 medulloblastoma would be of paramount importance for the identification and testing of new drugs for pediatric patients, tailored on the genetic condition of the patient itself. Recently, several studies have utilized next-generation sequencing technologies to map the genomic landscape of MB and to identify novel driver mutations[3–8]. A second-generation medulloblastoma subgrouping of Group 3/4 has led to the identification of eight subtypes with major clinicopathological and molecular features[9]. Group II, III, and V are at high clinical risk (5 years overall survival 41–58 months in retrospective series) and enriched for *c-MYC* amplification. Notably, the function and tumorigenicity of some oncogenes, such as *c-MYC*, have been tested ex vivo and also in vivo where its overexpression together with either Gfi1 overexpression or p53 loss of function, is sufficient to promote MB in mice[10–12]. The same gene combinations have been electroporated in utero to examine the susceptibility of distinct cerebellar progenitors to transformation[13]. Furthermore, p53 has been found to interact with c-Myc family in mouse models and human patients of Group 3 MB, suggesting their contribution to MB relapse[14].

In this study, we performed an in vivo screen by postnatal transfection of mouse cerebella to identify novel driver genes combinations able to induce Group 3 MB. Recently, human neuroepithelial stem cells (derived from iPSC) have been used to model MB. In particular, iPSC from patients with Gorlin syndrome have been differentiated in neuroepithelial stem cells, and it has been shown that these are able to induce MB upon orthotopic implantation in mice[15]. On the other hand, human forebrain organoids have been already used to test the function of putative genes involved in glioma[16–18], but cerebellar organoids have never been used to model human MB. Notably, human iPSC robustly differentiate into cerebellar progenitors and generate human cerebellar organoids that partially recapitulate the complexity of human cerebellum[19,20]. Here, we exploited human cerebellar organoids to develop reliable MB organoids and to validate our screen in human cells.

## Results

**In vivo transfection of cerebellar cells**. In order to uncover novel gene combinations that could be responsible for Group 3 MB development, we have reproduced in mice the genetic alterations found in human patients. In a first set of experiments, we tested our ability to target and transfect cerebellar cells in vivo. A mix of Venus-coding plasmid and jetPEI transfection reagent (Polyplus-transfection) were stereotaxically injected in vivo into newborn (P0) CD1 wild-type mice[21]. To obtain stable expression of Venus, we used PiggyBac system[22] that allows multiple insertions of Venus under control of CAG promoter. As shown in Fig. 1a–c, at 3, 7, and 23 days post injection (d.p.i.) we detected Venus-positive cells in external granule layer (EGL), internal granule layer (IGL), and in deep cerebellar nuclei (DCN). Interestingly, at 7 d.p.i. Venus is expressed in Sox9 (Fig. 1d) and Sox2 (Supplementary Fig. 1A) positive glial cells, in Olig2-positive oligodendrocytes (Fig. 1e), Barhl1-positive granule neuron progenitors (Supplementary Fig. 1B), few calbindin

positive Purkinje cells (Supplementary Fig. 1C), and NeuN-positive neurons (Supplementary Fig. 1D) in IGL. Therefore, with this strategy we have been able to perform stable transfection of glial cells, neurons, and progenitors in postnatal mouse cerebellum.

***Otx2* and *c-MYC* as novel MB driver genes**. To identify novel putative oncogenes/oncosuppressors combinations, we analyzed the list of patient-specific mutations identified in previous exome sequencing, microarray, and CNVs data of Group 3 MB[3–8]. We decided to focus on all the genetic alterations present in Group 3 MB patients with a frequency higher than 2% or genes that show differential expression (higher than 16-folds) compared with normal cerebellum[23] (see the Methods section). Based on this analysis, we created a list of gene combinations (Fig. 1f; Supplementary Fig. 1E) to be tested in mice for their ability to induce MB. To recapitulate the human gene amplification or overexpression, we used the PiggyBac system, which allows multiple insertions of the selected putative oncogene. On the other hand, we used CRISPR/Cas9-mediated loss-of-function approach to remove the selected putative oncosuppressors (Methods). Cas9 technology has been already used to model MB[21] and to screen genes involved in tumor growth and metastasis in mice[24,25].

Several gene combinations did not give rise to tumors but only to the formation of big clusters of cells with weak Venus expression 3 months post injection (Supplementary Fig. 1F). Since we never observed these cell clusters in control experiments (injection of Venus alone), we speculate that these could be dead or senescent cells due to either oncogenes expression or oncosuppressors deletion. None of the gene combinations, where putative oncosuppressors were silenced with Cas9 technology, led to tumor formation. This might be due to inefficient gene deletion or because of missense, nonsense, and frameshift mutations present in human patients are not efficiently recapitulated by our strategy. Among all the tested combinations, we observed reduced mice survival with *Gfi1* + *c-MYC* (GM) and *Otx2* + *c-MYC* (OM) genes overexpression (Fig. 1g) and formation of brain tumors (Fig. 1f, h, 2a, b). The *Gfi1* + *c-MYC* overexpression in postnatal cerebellar progenitors has been previously described as able to generate Group 3 MB in mice[11,26], therefore validating the efficiency of our method. As shown in Fig. 2a, b, GM and OM overexpression in mouse cerebellum induced tumors. The cells within the tumors express c-MYC (Supplementary Fig. 2A, B), Gfi1 (Fig. 2c) and Otx2 (Fig. 2d) and are in proliferation (Supplementary Fig. 2C, D). Notably, the tumors are NPR3 positive (Fig. 2e, f) and GFAP negative (Fig. 2g, h) such as human Group 3 MB[27,28]. In fact, NPR3 is a specific marker that is expressed in human Group 3 MB and is not present in the other MB subgroups[28], suggesting that our model could recapitulate human tumors. The histopathological and immunophenotypical analysis confirmed that tumors are Synaptophysin-positive MB and with cytoplasmic b-catenin, therefore, non-Wnt MB (Supplementary Fig. 2E–I). Notably, within the tumors, we observed Sox9/Venus double-positive cells, putative glial cells (Supplementary Fig. 3A, B), but also Venus-negative–Olig2-positive cells that could be infiltrated oligodendrocytes (Supplementary Fig. 3C, D). Furthermore, the tumors seemed to be Barhl1 negative, a marker for granule neuron progenitors and granule neurons (Supplementary Fig. 3E, F). Interestingly, in one mouse injected with OM we observed sacral metastasis (Venus and pH3 positive, Fig. 2i–l), suggesting that our new Group 3 mouse models can recapitulate the malignant phenotype of human MB. Interestingly, *OTX2* is overexpressed/amplified in human Group 3 MB and has been already found to be required for the tumorigenesis of MB cell lines[29–31].

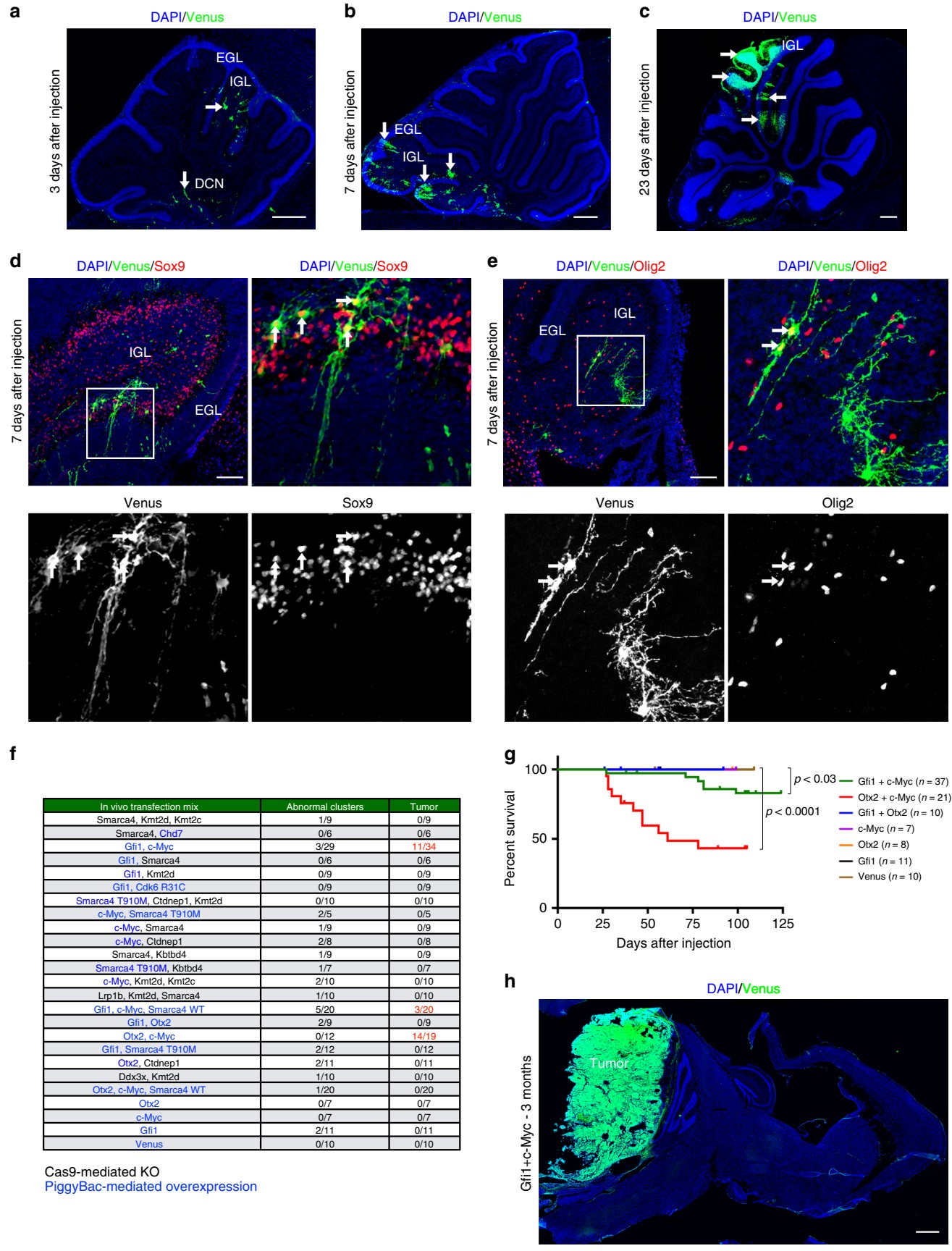

**Fig. 1 In vivo transfection of mouse cerebellum with different gene combinations. a–c** Confocal images of DAPI staining and GFP immunofluorescence (Venus) of sagittal brain section of CD1 mouse 3 (**a**), 7 (**b**), 23 (**c**) days after transfection with pPBase and pPBVenus at P0. Arrows point to Venus-positive cells. **d** Confocal images of DAPI staining and immunofluorescence for GFP (Venus) and Sox9 of sagittal brain section of CD1 mouse 7 days after transfection with pPBase and pPBVenus at P0. **e** Confocal images of DAPI staining and immunofluorescence for GFP (Venus) and Olig2 of sagittal brain section of CD1 mouse 7 days after transfection with pPBase and pPBVenus at P0. The white square in (**d**, **e**) marks the region shown at higher magnification. **f** List of combinations of putative oncogenes and putative oncosuppressors transfected in CD1 mice at P0. Putative oncogenes are reported in blue, putative oncosuppressors are reported in black. The number of mice displaying abnormal clusters and tumors are reported. **g** Kaplan–Meier survival curve of mice injected at P0 with different gene combinations. **h** DAPI staining and GFP immunofluorescence (Venus) of sagittal brain section of CD1 mouse 3 months after transfection with pPBase + pPBMyc + pPBGfi1 + pPBVenus at P0. Scale bars 250 μm (**a**), 500 μm (**b**, **c**), 100 μm in (**d**, **e**), 1 mm in (**h**).

Here, we have demonstrated that Otx2 is able to induce MB in vivo from mouse cerebellar progenitors that have not been manipulated outside the mouse brain. Our data indicate that concomitant overexpression of Otx2 and c-MYC originate MB in vivo. The two genes alone are not able to induce MB, suggesting that cooperation between the two transcription factors is required for tumor formation. Notably, the combination of *Gfi1* and *Otx2* overexpression does not induce MB (Fig. 1f), therefore suggesting that *c-MYC* alteration is required to transform cerebellar cells.

**SMARCA4 blocks *Otx2*/*c-MYC*-induced Group 3 MB**. To identify a druggable signaling pathway representing a possible new therapeutic alternative to block Group 3 MB, we analyzed the sequencing data of human patients harboring *OTX2* and *c-MYC* overexpression. Among all the putative oncosuppressors altered in Group 3 MB, *SMARCA4* is the most frequently mutated (harboring missense mutations in helicase domains), highlighting its possible role in MB formation. Notably, in Group 3 MB patients (that have high levels of OTX2 and c-MYC), there are variable levels of *SMARCA4* expression that are lower than in human neural stem cells (Supplementary Fig. 4A, B). Furthermore, endogenous Smarca4 expression is downregulated in our OM-induced MB mouse models as compared with the GM tumors (Fig. 3a–c). Based on this evidence, we speculate that our mouse model could represent patients expressing high levels of OTX2/c-MYC and low levels of SMARCA4. Therefore, we expressed high levels of SMARCA4 with the intent of inhibiting OM-induced MB generation. Interestingly, 20/20 mice injected with both OM and SMARCA4 overexpression vectors were still alive after 3 months and none of them developed MB (Fig. 3d). On the other hand, SMARCA4 overexpression partially reduced the incidence of MB in GM animals and 3/20 mice developed MB with no influence on their survival (Fig. 3e). These results suggest that different Group 3 MB tumors might show different sensitivity to *SMARCA4* deregulation.

To better characterize the effects of SMARCA4, we analyzed the cerebellum of mice injected with OM with and without SMARCA4 overexpression, 10 days after injection. As shown in Fig. 3f, OM overexpression led to formation of Ki67/Venus-positive cells clusters with few β3-tubulin and GFAP-positive cells (Fig. 3g, h). Notably, when SMARCA4 was co-overexpressed, there was no formation of cell clusters and we identified only few Ki67/Venus double-positive cells (Fig. 3i). Furthermore, several Venus-positive cells expressed β3-tubulin and GFAP (Fig. 3j, k). We observed that Otx2 and c-MYC proteins are still present in transfected cerebellum with SMARCA4 co-overexpression (Supplementary Fig. 4C–Q), suggesting that SMARCA4 does not affect their expression, but Otx2/c-MYC oncogenic-mediated activity. Finally, we tested if SMARCA4 overexpression could lead to abnormal levels of other proteins part of the SWI/SNF complex and be responsible for the phenotype observed. As shown in Supplementary Fig. 4R, SMARCA4 overexpression in human

cerebellar progenitors that have a gene expression pattern resembling hindbrain fate (iPSC-derived AF22 cells)[32,33] seems to have no effect on the SWI/SNF complex proteins expression.

**Human cerebellar organoids as a reliable model for Group 3 MB**. To generate an organoid-derived model of human Group 3 MB, we tested the function of GM and OM combinations in human cerebellar organoids. Notably, Muguruma et al.[19,20] clearly demonstrated that hESC/iPSC aggregates are steered to differentiate into cerebellar progenitors and neurons. In fact, they were able to induce progenitors to self-form neuroepithelial structures that mimic early cerebellar plate, composed by cerebellar progenitors, cerebellar neurons (interneurons, Purkinje cells, and granule neurons) and glial cells. We recapitulated Muguruma protocol and we electroporated PiggyBac vector expressing Venus into human cerebellar organoids (Fig. 4a). As shown in Fig. 4b and Supplementary Fig. 5A, B, we were able to electroporate human cerebellar organoids at day 35 of differentiation, when almost all cerebellar progenitors are present[20] and to deliver DNA into intact 3D human cerebellar organoids (Supplementary Fig. 5B). In order to confirm whether the electroporation procedure induced any change in the transcriptome of the cerebellar organoids, we performed an analysis of the differentially expressed genes (DEGs). As seen in Supplementary Data 1, the number of DEGs is very small (upregulated in treated $n = 26$; downregulated in treated $n = 7$; log fold change > 1, count per million average expression > 3, Benjamini–Hochberg adjusted the $p$-value ≤ 0.05). Moreover, the functional analysis in those small gene subsets indicated biological terms enriched in extracellular matrix/cell adhesion and in protein folding (for details see Supplementary Data 1). We then proceeded to validate our patient-specific screen results in the human organoids by electroporating them at day 35 with GM and OM. As shown in Fig. 4c, d, we observed the formation of small buds of Venus-positive cells in GM and OM electroporated organoids after 25 days, but not in control organoids electroporated with Venus only (Fig. 4b). Notably, the GM and OM organoids showed increased number of PCNA-positive cells within the Venus-positive cells compared with control organoids (Fig. 4e, g; Supplementary Fig. 5C, D). Furthermore, OM organoids (within the Venus-positive cells) had less β3-tubulin-positive cells compared with control cells (Fig. 4f, g). Taken together, these data suggest that GM and OM combinations induce overproliferation of human cerebellar progenitors and impair their differentiation (Fig. 4g). Notably, we observed an increase in Sox9-positive cells that could indicate an increase in glial cells or early cerebellar progenitors, while we did not observe any change in the number of Skor2-positive Purkinje cell precursors (Fig. 4g; Supplementary Fig. 5E–H). To understand how GM and OM combinations modify the proliferation of cerebellar progenitors, we analyzed the organoids at an earlier time point (5 days after electroporation). As shown in Fig. 4h, at this stage we observed an increase in PCNA-positive cells possibly due to the mitotic activity of GM

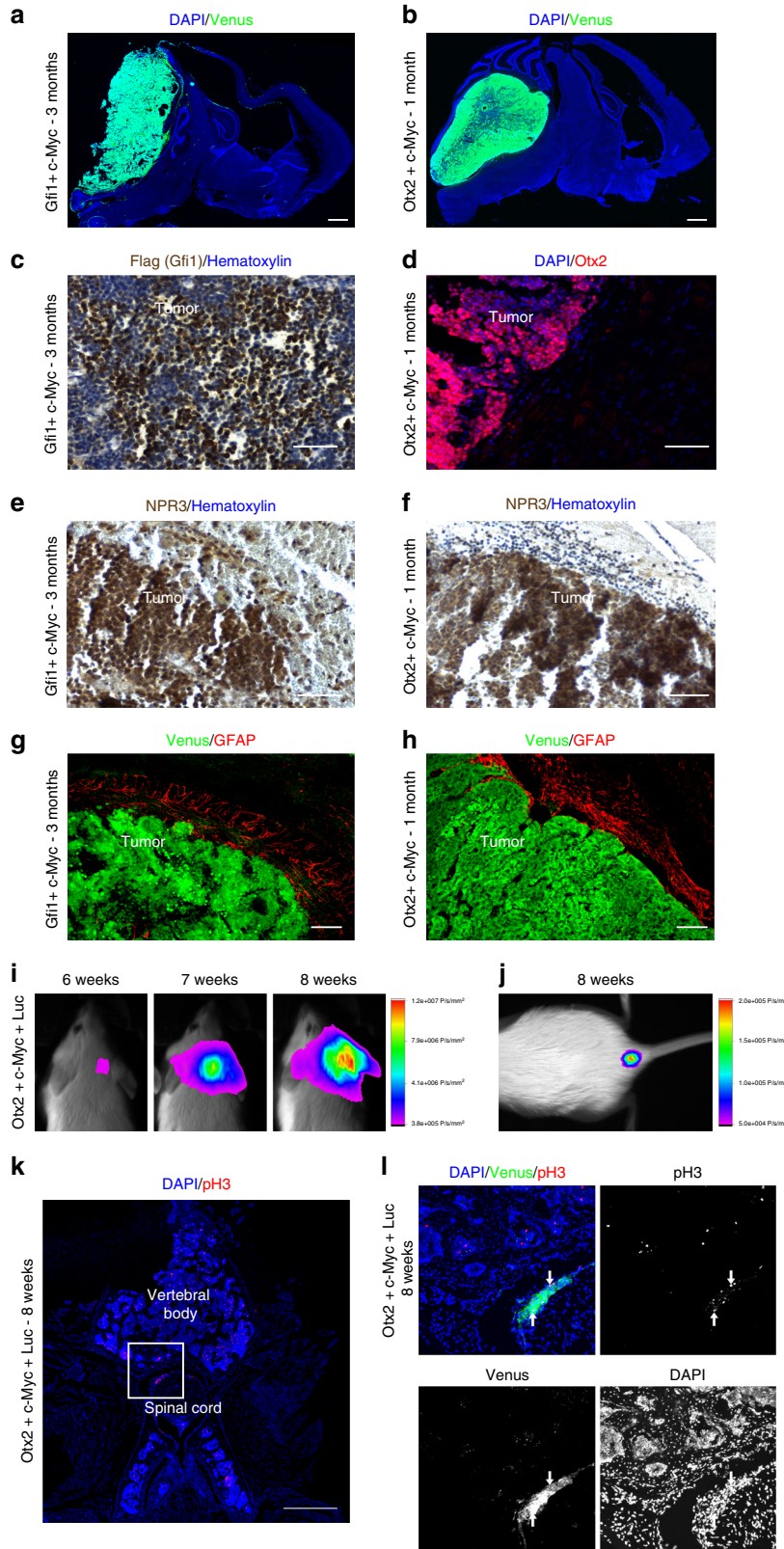

and OM combinations. Since our previous data in mice suggest that SMARCA4 has an effect in reducing OM tumorigenesis, we tested its effects in human cerebellar organoids. As shown in Supplementary Fig. 5I, J, SMARCA4 overexpression is able to reduce OM-induced proliferation in human cerebellar organoids, confirming that SMARCA4 might have a role in reducing OM-induced Group 3 MB. Notably, SMARCA4 overexpression was not able to reduce GM-induced proliferation in human cerebellar organoids, confirming our findings in mice (Supplementary Fig. 5K). To test whether organoid-derived model of human Group 3 MB is able to induce cancer in vivo, we injected them in Foxn1[nu] (nude) mice (Fig. 5a). We observed the formation of MB

**Fig. 2 In vivo transfection of cerebellar cells with *Gfi1/c-MYC* and *Otx2/c-MYC* induces Group 3 MB. a** DAPI staining and GFP immunofluorescence of CD1 mouse brain section 3 months after transfection with pPBase + pPBMyc + pPBGfi1 + pPBVenus at P0. **b** DAPI and GFP immunofluorescence of CD1 mouse brain section 1 month after transfection with pPBase + pPBMyc + pPBOtx2 + pPBVenus at P0. **c** Hematoxylin and Flag (Gfi1) immunohistochemistry of CD1 mouse tumors after transfection with pPBase + pPBMyc + pPBGfi1 + pPBVenus at P0. **d** DAPI and Otx2 immunofluorescence of CD1 mouse brain section 1 month after transfection with pPBase + pPBMyc + pPBOtx2 + pPBVenus at P0. **e, f** Hematoxylin and NPR3 immunohistochemistry of CD1 mice tumors after transfection with pPBase + pPBMyc + pPBGfi1 + pPBVenus at P0 (**e**) and with pPBase + pPBMyc + pPBOtx2 + pPBVenus at P0 (**f**). **g, h** GFP and GFAP immunofluorescence of tumors in CD1 mice after transfection with pPBase + pPBMyc + pPBGfi1 + pPBVenus at P0 (**g**) and with pPBase + pPBMyc + pPBOtx2 + pPBVenus at P0 (**h**). **i** In vivo bioluminescent imaging of the upper body of a CD1 mouse transfected with pPBase + pPBMyc + pPBOtx2 + pPBVenus at P0, with a plasmid encoding firefly luciferase (pPBLuc). Imaging was performed weekly from the sixth week after transfection. **j** In vivo bioluminescent imaging of the lumbosacral region of a CD1 mouse 8 weeks after transfection with pPBase + pPBMyc + pPBOtx2 + pPBVenus + pPBLuc at P0. The luminescence signal is expressed in photons/sec/mm2. **k** DAPI and immunofluorescence for pH3 of transversal section of the sacral portion of the spine of a CD1 mouse 8 weeks after transfection with pPBase + pPBMyc + pPBOtx2 + pPBVenus + pPBLuc at P0. **l** DAPI and immunofluorescence for GFP and pH3 of sacral transversal section of the spine of a CD1 mouse 8 weeks after transfection with pPBase + pPBMyc + pPBOtx2 + pPBVenus + pPBLuc at P0. **l** is the higher magnification of the white square in (**k**). Scale bars 1 mm in (**a**, **b**), 100 μm in (**c–h**), 500 μm in (**k**).

in nude mice injected with GM (11/11)- and OM (7/7)-modified organoids, but not with Venus (0/6) (Fig. 5b; Supplementary Figs. 6A, C, E, 7, 8). The Venus-positive cells (in GM- and OM-modified organoids injected in mice) are pH3 and PCNA-positive, suggesting that the tumor is still growing several days after injection (GM:72 days, OM:30 days, Fig. 5c; Supplementary Fig. 6A–F). Notably, the tumors originating from OM-injected organoids are Otx2, NPR3 positive and GFAP, Olig2, Barhl1 negative (Fig. 5d; Supplementary Fig. 7A–D) similarly to the tumors induced by OM injection in CD1 mice (Fig. 2d, f, h; Supplementary Fig. 3D, F) and to human Group 3 MB[27,28]. This confirms that Gfi1, Otx2 and c-MYC are sufficient to induce MB once overexpressed into human cerebellar organoids. To classify our new Group 3 MB derived from human organoid, we analyzed the global DNA methylation profile of both OM- and GM-injected organoids and compared them with those derived from MB patients, diagnosed and treated at the Ospedale Pediatrico Bambino Gesù (OPBG Rome). This approach has been recently developed and used for clinical decision-making[34,35]. The clinical and pathological features of the 36 primary MBs we analyzed are summarized in Supplementary Data 2. Injected organoid samples displayed a global methylation profile close to those of Group 3 MBs, as evidenced by both multidimensional scaling analysis performed on the 1000 most variable islands in the cohort (Fig. 5e), and hierarchical clustering analysis (Fig. 5f; Supplementary Fig. 8D) performed on the reduced 48 CpG islands signature that better characterize MB subgroups[35]. The methylation data files from our organoids were also run through the brain tumor classifier (https://www.molecularneuropathology.org/mnp)[34]. All samples were classified in the Methylation class family Medulloblastoma G3 and G4 (subclass Group 3) with a score > 0.3 (Supplementary Data 3). Of note, by exploiting the Medulloblastoma classifier Group 3/4 v1.0 that takes into account the new consensus on the second-generation molecular sub-grouping of medulloblastoma[9], GM samples were classified as Subtype II, high-risk G3 tumors, while OM samples as Subtype IV, standard-risk G3 tumors. These data highlight the differences obtained by manipulating distinct genes in our models. Therefore, our organoid-based model for Group 3 MB could be used as a reliable tool for human MB modeling.

**Mutant SMARCA4 represses wild-type SMARCA4 functions.** To better characterize the mechanisms underlying SMARCA4 in MB development, we have overexpressed OM and *SMARCA4* in human cerebellar progenitors (AF22 cells) and we checked the expression of several downstream genes of OTX2, c-MYC, and SMARCA4 (Fig. 6a, b). Among the different analyzed genes, we found that OM affects *CDKN2B* and *CRABP1* expression and that

SMARCA4 is able to rescue these effects (Fig. 6c). Notably, SMARCA4 loss of function does not influence *CDKN2B* and *CRABP1* modulation by OM (Fig. 6d, e). In line with our results, p15[INK4b] protein (encoded by *CDKN2B* gene) is a well-known oncosuppressor whose modulation by c-MYC and SMARCA4 has already been suggested in cell lines[36–39]. CRABP1 protein binds to retinoic acid and helps to transport it into cells. Its role in human carcinogenesis is poorly understood, but high CRABP1 expression levels are associated with poor patient prognosis, high tumor grade in breast cancer[40] and CRABP1 protein modulates cell cycle progression and apoptosis induction in mouse and human cell lines[41]. In Group 3 MB patients, *SMARCA4* gene presents several missense mutations (12 out of 131 patients) and the most common SMARCA4 mutation T910M (3 out of 12) has been already characterized in mouse fibroblast, human cell lines, and SMARCA4-deficient cell lines[42–44], but not in cerebellar progenitors. Furthermore, SMARCA4 T910M protein has been found normally incorporated into the BAF complex, but its ATPase activity is highly compromised[42–44]. Since SMARCA4 T910M mutation is mainly present in heterozygosity, to analyze its function, we co-overexpressed (at comparable levels, Fig. 6b) the wild-type and mutated form with OM. Interestingly, SMARCA4 T910M co-overexpression blocks the SMARCA4 wild-type effects on *CDKN2B* and *CRABP1* gene expression (Fig. 6c). To further characterize SMARCA4 T910M functions, we analyzed its effects in vivo. As shown in Fig. 6f, g, SMARCA4 wt is able to block OM-induced MB while SMARCA4 T910M overexpression is able to counteract SMARCA4 wt effects. We further analyzed SMARCA4 T910M effects in human cerebellar organoids and as shown in Fig. 6h, it is able to block the SMARCA4 wild-type effects also in this system. Taken together, these data suggest that in patients harboring heterozygotes *SMARCA4* missense mutations, SMARCA4 T910M represses SMARCA4 wild-type functions and acts as a dominant-negative.

**EZH2 inhibition reduces Group 3 MB growth.** SMARCA4 has an antagonistic relationship with histone methyltransferase EZH2 (part of the PRC2 complex), which indeed is essential for survival and growth of SMARCA4-deficient cancer cells[45,46]. Therefore, in search of novel molecules to be used to treat OM MB, we focused on a nonspecific inhibitor of EZH2, 3-deazaneplanocin A (DZNep). Treatment with DZNep induces significant antitumor activity in various cancer types, corresponding to inhibition of PRC2 and removal of H3K27me3 marks[47–49]. First, we performed ex vivo culture of cancer cells from mouse tumors arisen upon OM injection. Ezh2 inhibition blocked OM-induced cancer cells growth, DZNep reduced the size of the ex vivo cultures of OM-derived tumor spheroids (Supplementary Fig. 9A–C). To

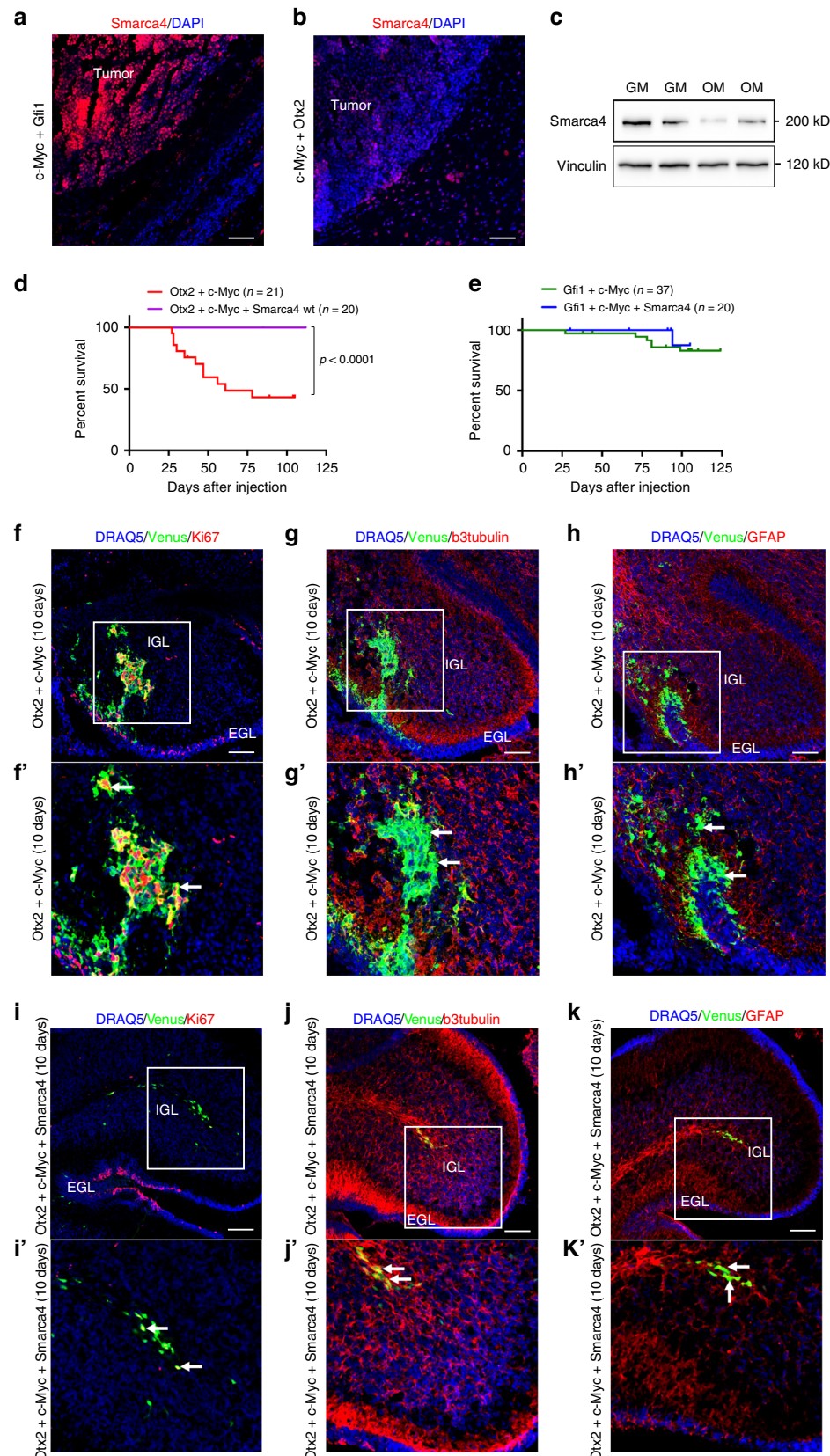

compare DZNep effects with other treatments, we used also Panobinostat (LBH589) a histone deacetylase (HDAC) inhibitor that inhibits murine and human MYC-driven MB in vitro and in vivo[50,51]. As shown in Supplementary Fig. 9A–C, Panobinostat (LBH589) had a significantly lower effect on OM-derived tumor

spheroids growth, as compared with DZNep. To further test DZNep functions in human cells, we used our OM organoid-derived model of human Group 3 MB. As shown in Supplementary Fig. 9D–F, DZNep is able to block the growth of OM electroporated cells due to the decreased number of GFP/PCNA-

**Fig. 3 SMARCA4 represses *Otx2/c-MYC* induced Group 3 medulloblastoma. a** DAPI staining and Smarca4 immunofluorescence of sagittal brain tumor section of CD1 mouse 3 months after transfection with pPBase + pPBMyc + pPBGfi1 + pPBVenus at P0. **b** DAPI staining and Smarca4 immunofluorescence of sagittal brain tumor section of CD1 mouse 1 month after transfection with pPBase + pPBMyc + pPBOtx2 + pPBVenus at P0. **c** Western blot of brain tumors of CD1 mice after transfection with pPBase + pPBMyc + pPBGfi1 + pPBVenus (GM) and pPBase + pPBMyc + pPBOtx2 + pPBVenus (OM). **d** Kaplan–Meier survival curve of mice injected at P0 with Otx2 + c-Myc and Otx2 + c-Myc + Smarca4. **e** Kaplan–Meier survival curve of mice injected at P0 with Gfi1 + c-Myc and Gfi1 + c-Myc + Smarca4. **f–h** Confocal images of GFP (Venus) and Ki67 (**f**), β3-tubulin (**g**), GFAP (**h**) immunofluorescence of transfected cell clusters in CD1 mouse 10 days after transfection with pPBase + pPBMyc + pPBOtx2 + pPBVenus at P0. The white squares in (**f, g, h**) mark the region shown at higher magnification in (**f′, g′, h′**). **i–k** Confocal images of GFP (Venus) and Ki67 (**i**), β3-tubulin (**j**), GFAP (**k**) immunofluorescence of transfected cell clusters in CD1 mouse 10 days after transfection with pPBase + pPBMyc + pPBOtx2 + pPBSmarca4 + pPBVenus at P0. The white squares in (**i, j, k**) mark the region shown at higher magnification in (**i′, j′, k′**). Arrows point to double-positive cells. Scale bars 100 μm.

positive cells. We have then tested whether EZH2 inhibitor (DZNep) induces apoptosis and/or cell death in tumor spheroid. As shown in Supplementary Fig. 10A, B, 24h after DZNep treatment almost 90% the cells inside tumor spheroids were dead or in apoptosis. Furthermore, we observed an increase in Cleaved Caspase-3-positive cells with DZNep treatment compared with control (Supplementary Fig. 10C, D) and a decrease in the number of pH3-positive cells (Supplementary Fig. 10E, F). Since DZNep is not a specific inhibitor only for EZH2, we tested (in mouse OM-derived tumor spheroids) other two inhibitors that more specifically repress EZH2 functions, Tazemetostat (EPZ-6438) and GSK-126[52,53]. As shown in Fig. 7a, b upon 24 h of treatment with Tazemetostat and GSK-126, we did not observe increased apoptosis or differences in cell cycle compared with control. On the other hand, upon 3 days of Tazemetostat treatment, we observed an increased number of cells either dead or in late apoptosis (Fig. 7c, d), reduced number of cells in cell cycle compared with control (Fig. 7e, f). We further tested Tazemetostat and GSK-126 treatments in OM-derived MB produced in human organoids. As shown in Fig. 7g, h, we observed an increase in the number of Cleaved Caspase3 and a decrease in PCNA-positive cells in organoids treated for 5 days with Tazemetostat compared with control. In conclusion, the analysis performed in our MB models suggests that EZH2 inhibition with Tazemetostat might represent a suitable therapeutical strategy for Group 3 MB patients with high levels of OTX2 and c-MYC.

## Discussion

Human Group 3 MB is one of the most aggressive MB subgroup and is characterized by high c-MYC levels. To recapitulate Group 3 MB tumors, several research groups have overexpressed c-Myc together with either Gfi1 or dominant-negative p53[10–12,26]. Those modifications were sufficient to promote Group 3 MB in mice in vivo and in isolated CD133-positive progenitors from postnatal murine cerebellum. Moreover, drug screens have been performed in Group 3 MB mouse models and patient-derived xenografts (PDX) that led to the identification of molecules able to increase mouse survival[51,54]. However, Group 3 and MB, in general, represents a very heterogeneous family of tumors that requires different therapeutic approaches to obtain reliable results and lower the side effects. Up to now, no patient-specific MB models exist and specific therapeutic strategies for the different Group 3 MB patients are lacking. We performed an in vivo screen to identify new cancer driver genes starting from published genome-wide analyses of MB patients. Our analysis led to the identification of *Otx2/c-MYC* (OM) as a novel driver gene combination required for tumorigenesis. OTX2 is highly expressed in non-SHH MB subgroups, and *OTX2* locus is amplified and/or overexpressed in a subset of Group 3 and Group 4 tumors. Previously, it has been shown that OTX2 is required for human MB cell lines growth, but its capacity of inducing MB from cerebellar cells in vivo has never been tested. Furthermore, OTX2 downstream genes *NRL* and *CRX* are required for the MB formation in mice

using Group 3 PDX[55]. Here, we demonstrated that Otx2 overexpression is necessary for c-MYC tumorigenesis in mouse cerebellar cells in vivo. To confirm the relevance of our data for human tumorigenesis, we used human cerebellar organoids. We generated human iPSC-derived cancer organoids with GM and OM overexpression mimicking human Group 3 MB genetic alteration. Indeed, the use of DNA methylation signature[34] in combination with Group 3-specific markers analysis indicates that our organoid-based MB model recapitulates several features of human Group 3 MB. Of note, exploiting the MB classifier group 3/4[9], the GM cancer organoids were classified as Subtype II, high-risk G3 tumors, while OM cancer organoids as Subtype IV, standard-risk G3 tumors. These data highlight the differences obtained by manipulating distinct genes in our models. Therefore, our organoid-based model for Group 3 MB can be used as reliable tool to generate novel patient-specific cancer models. Moreover, our data show that OM oncogenic properties are inhibited by the chromatin modifier SMARCA4, a gene also found mutated (putative loss of function) in Group 3 MB that has been already described as involved in cerebellum development and SHH MB[56,57]. Since SMARCA4 is not able to block the oncogenic role of GM combination in Group 3 MB formation, we speculate that patients with Group 3 tumors should be treated differently, depending on their mutations. As a core subunit of the SWI/SNF chromatin remodeling complex, Smarca4 contains an ATPase domain that provides the enzymatic activity required to remodel chromatin structures and regulate transcription[58,59]. Mutations that inactivate SWI/SNF subunits are found in ~20% of human cancers, suggesting that a functional complex might be required to prevent tumor formation in several tissues[58,59]. *SMARCA4* gene presents several missense mutations in Group 3 MB patients, and the most common SMARCA4 mutation is the T910M. SMARCA4 T910M protein has been found incorporated normally into the BAF complex, but leads to highly compromised ATPase activity[42,43]. Notably, SMARCA4 is involved in decatenating newly replicated sister chromatids, a step required for proper chromosome segregation during mitosis[42]. Failure in this process could lead to aneuploidy that is common in MB and ranges from the partial gain or loss of single chromosomes to full tetraploidy[3,42]. However, it has been published that the relative rate of tetraploidy of *SMARCA4* mutant MB was similar to that of wild-type *SMARCA4* MB[3]. Therefore, SMARCA4 could have also other mechanisms by which it is involved in MB development. Since SMARCA4 T910M mutation is mainly present in heterozygosity in human Group 3 MB, we analyzed its function in co-overexpression with wild-type SMARCA4 in human neuroepithelial cells, in vivo and in human cerebellar organoids. Our data show that SMARCA4 T910M is able to reduce SMARCA4 wild-type functions on *CDKN2B* and *CRABP1* expression and restores the Otx2/c-MYC oncogenic functions. Based on this evidence, we speculate that mutant SMARCA4 T910M could act as dominant-negative and block the wild-type form. Supporting this hypothesis, some other SMARCA4 mutants act as dominant-negative in

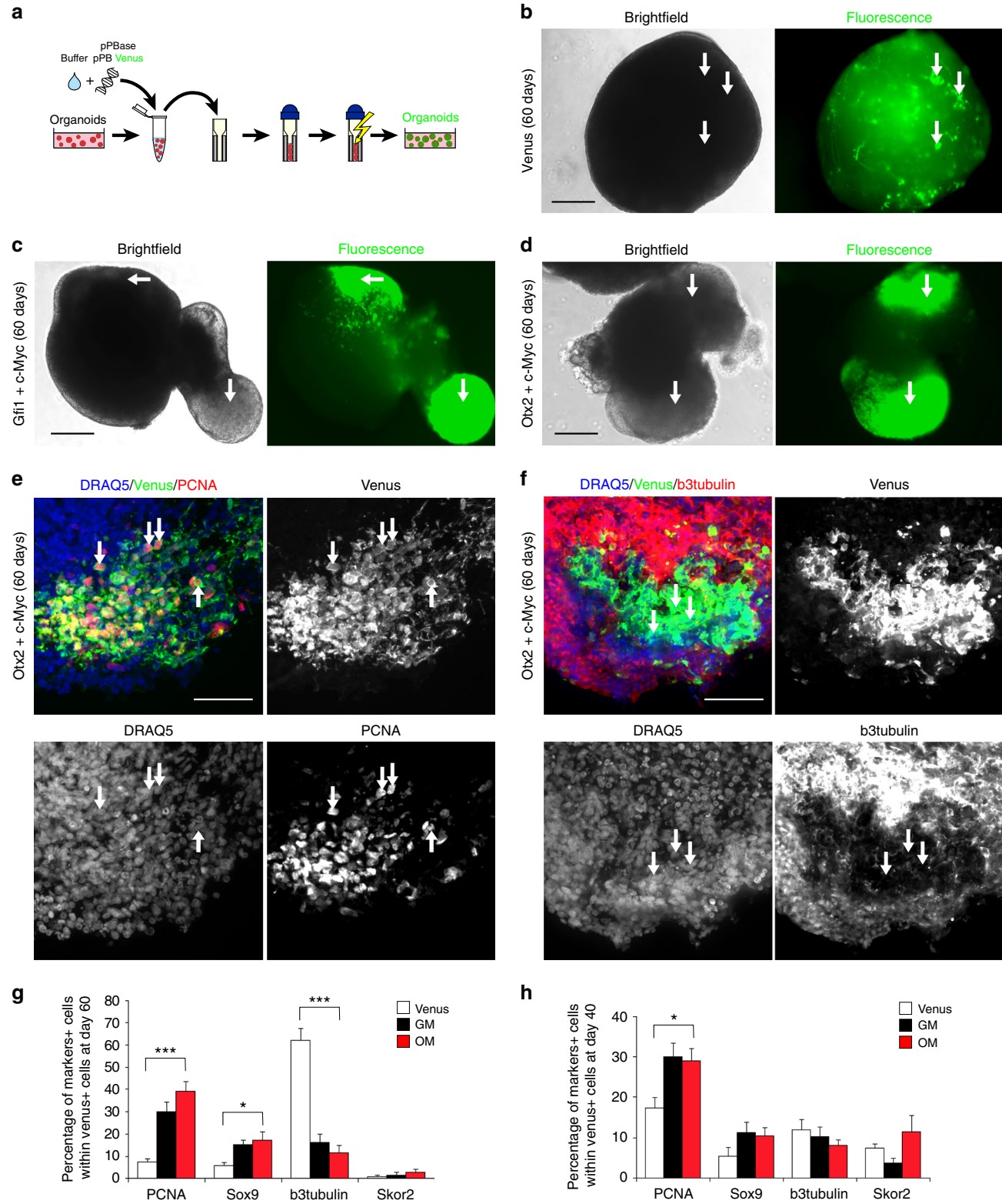

human cell lines[60], but their function in vivo (during MB development) has not been investigated. Finally, a recent study clarified that SMARCA4 T910M contributes to the targeting of SWI/SNF complexes on chromatin in ovarian carcinoma cells and that could influence *CDKN2B* expression[44]. This evidence suggests that SMARCA4 T910M might play similar roles in MB and ovarian carcinoma development. Interestingly, SMARCA4 has been previously found to be partially required for SHH

MB development, suggesting different roles of SMARCA4 depending on the MB subgroups and on the genetic alteration that are involved in tumor generation[57]. The SWI/SNF complex has an antagonistic relationship with polycomb repressive complex 2 (PRC2) in several tumors[45,46]. Indeed, histone methyltransferase EZH2 (part of the PRC2 complex) is essential for survival and growth of SMARCA4-deficient cancer cells[45]. Furthermore, EZH2 and polycomb genes are strongly upregulated in

**Fig. 4 Cerebellar organoids electroporation with *Gfi1/c-MYC* and *Otx2/c-MYC* induces overproliferation. a** Schematic representation of organoids electroporation. **b** Brightfield and fluorescence images of cerebellar organoids at day 60 electroporated at day 35 with pPBVenus. **c** Brightfield and fluorescence images of cerebellar organoids at day 60 electroporated at day 35 with pPBase + pPBMyc + pPBGfi1 + pPBVenus. **d** Brightfield and fluorescence images of cerebellar organoids at day 60 electroporated at day 35 with pPBase + pPBMyc + pPBOtx2 + pPBVenus. Arrows in (**b**, **c**, **d**) indicate Venus-positive cells. **e** Confocal images of GFP (Venus) and PCNA immunofluorescence of cerebellar organoids at day 60 electroporated at day 35 with pPBase + pPBVenus + pPBMyc and pPBOtx2. Arrows indicate double-positive cells. **f** Confocal images of GFP (Venus) and β3-tubulin immunofluorescence of cerebellar organoids at day 60 electroporated at day 35 with pPBase + pPBMyc + pPBOtx2 + pPBVenus. Arrows indicate β3-tubulin-negative cells. **g** Quantification of cerebellar organoids at day 60, electroporated at day 35 with either pPBVenus or pPBase + pPBMyc + pPBOtx2 + pPBVenus (OM) or pPBase + pPBMyc + pPBGfi1 + pPBVenus (GM). $n = 6$ biologically independent organoids. **h** Quantification of cerebellar organoids at day 40, electroporated at day 35 with either pPBVenus or pPBase + pPBMyc + pPBOtx2 + pPBVenus (OM) or pPBase + pPBMyc + pPBGfi1 + pPBVenus (GM). $n = 6$ biologically independent organoids. Error bars in (**g**, **e**) represent standard error of the mean. Scale bars 250 μm in (**b–d**),100 μm in (**e**, **f**). Paired Student's *t* test, two tails. *$p$-value < 0.05, **$p$-value < 0.01. ***$p$-value < 0.001.

medulloblastoma compared with the normal cerebellum, especially in Group 3 and 4 tumors[8,61]. On the other hand, inactivation of Ezh2 accelerates tumor initiation in a mouse model of Group 3 MB induced by Gfi1 and c-Myc, suggesting a different role for these players in different subset of MB[26]. These results could explain why, in our hands, SMARCA4 overexpression reduces *Otx2/c-MYC* MB, but does not affect *Gfi1/c-MYC* tumorigenesis. Taken together, these studies suggest a possible link between SMARCA4 activity and EZH2 inhibition. 3-deazaneplanocin A (DZNep) is the first nonspecific EZH2 inhibitor that has been widely used and induces significant antitumor activity in various cancer types, resulting into inhibition of PRC2 and removal of H3K27me3 marks[62]. Indeed, the EZH2/PRC2 complex is involved in H3K27 methylation, and OTX2 has been found to sustain H3K27 trimethylation in human MB cell lines[61]. Here we show that DZNep reduces *Otx2/c-MYC* cancer cells growth ex vivo highlighting a possible therapeutic function of EZH2 inhibition. Moreover, EZH2 inhibitor also reduces OM cancer cells growth in human cerebellar organoids, suggesting that histone methyltransferases represent a promising therapeutic target in human cells. In spite of potentially promising results, DZNep has a very short plasma half-life, confers nonspecific inhibition of histone methylation and is toxic in animal models[63]. High-throughput biochemical screens have developed potent EZH2 inhibitors, namely GSK-126 and Tazemetostat (EPZ-6438)[52,53,64], that recently moved to clinical trial (PhaseI for lymphoma[48,65]). Tazemetostat is a potent and highly selective EZH2 inhibitor that has shown antitumor activity in vitro and in SMARCA4-negative malignant rhabdoid tumor of the ovary[52,66]. Interestingly, a recent Phase 2 Clinical Trial is currently determining the therapeutic effects of Tazemetostat in patients with solid tumors and non-Hodgkin lymphoma that do not respond to standard treatments and have *EZH2, SMARCB1,* or *SMARCA4* gene mutations (NCT03213665)[67]. Based on our data (Tazemetostat effects on mouse and human MB), we speculate that Tazemetostat could be a valuable treatment in Group 3 MB patients with high OTX2/c-MYC levels and low levels of SMARCA4 or *SMARCA4* mutations. This would suggest to recruit patients affected by a specific MB subtype during some ongoing or future clinical trials. The described results prove the usefulness of human organoids to investigate molecular mechanisms underlying cancer development. More importantly, the creation of specific MB subgroup organoids demonstrates how the system can be used to model MB taking into consideration its molecular stratification. All together our data will pave the road to use organoids-based models for a more tailored drug screen and therapy.

## Methods

**Plasmids**. The plasmid encoding an hyperactive form of the piggyBac transposase (pCMV HAhyPBase, pPBase) was a gift from https://www.sanger.ac.uk/form/

Sanger_CloneRequests[68]. The piggyBac donor plasmid pPB CAG c-MYC was a gift from https://www.sanger.ac.uk/form/Sanger_CloneRequests[69]. This plasmid was used as piggyBac donor backbone to clone by PCR other coding sequences, replacing cMyc coding sequence. Venus was amplified from pSCV2[70], to generate pPB CAG Venus plasmid (pPBVenus). mCherry coding sequence was amplified to generate pPB CAG mCherry. The firefly Luciferase coding sequence was cloned from pGL3 (Promega) into pPB CAG. Murine Gfi1 cDNA was amplified by PCR from Gfi1 NGFR (Addgene Plasmid #44630) and tagged by inserting in frame the FLAG-tag sequence at the 3′ end of the coding sequence. FLAG-tagged Gfi1 cDNA (Addgene #44630) was subcloned into the piggyBac donor backbone together with a IRES-GFP cassette, generating the plasmid pPB CAG Gfi1:FLAG-IRES-GFP. The piggyBac donor plasmids pPB CAG Cdk6 R31C-IRES-GFP, pPB CAG Otx2-IRES-GFP, and pPB CAG SMARCA4-IRES-GFP were generated by substituting Gfi1: FLAG with murine Cdk6 R31C, murine Otx2 or human SMARCA4 cDNAs, which were amplified by PCR from pcDNA3.1mouse cdk6 R31C (Addgene Plasmid #75171) and pBS hBRG1(Smarca4) (a gift from Anthony Imbalzano), respectively. A single-nucleotide mutation (C2729T) was introduced in the human SMARCA4 sequence to generate the missense mutation T910M using a one-step PCR method. pPB CAG SMARCA4-Venus and pPB CAG Smarca4 T910M-Venus plasmids encoding for wild-type and mutant SMARCA4-Venus fusion proteins were generated by assembling Venus-coding sequence in frame with wild-type or mutant SMARCA4 coding sequences. The SMARCA4-Venus and SMARCA4 T910M-Venus constructs were cloned into into pPB CAG backbone without a IRES-GFP cassette. The PiggyBac donor plasmid pPB CAG Otx2 was generated by PCR amplification of Otx2 coding sequence from pPB TetO-Otx2 UBC-rtTA and cloned into into pPB CAG backbone without a IRES-GFP cassette.

Three different double-stranded oligonucleotides coding for human *SMARCA4* 3′-UTR shRNA (target sequence shRNA1: 5′-GCTGTAGGACTGTTTGTGA-3′; target sequence shRNA2: 5′- CGGGTAGCAGCAGATGTAG-3′; target sequence shRNA3: 5′-TTGGGGAACACACGATACC-3′) and control shRNA (target sequence: 5′-ACTACCGTTGTTATAGGTG-3′) were cloned downstream of the U6 promoter into the pSCV2 plasmid according to the pSilencer instructions from Ambion.

The pSpCas9(BB)-2A-GFP (PX458) plasmids bearing three different sgRNAs targeting murine *Smarca4* gene were purchased from GenScript. All the other sgRNAs were cloned into pSpCas9 (BB)-2A-GFP (PX458) (Addgene plasmid #48138). The sequences of the sgRNAs used are listed below:

| gRNA | Sequence | |
|------|----------|---|
| Smarca4 g1 | CCACCCTCAGTGTCCGCCAC | Genscript |
| Smarca4 g2 | AGGCATGTTCAGAGCCGCCG | Genscript |
| Smarca4 g3 | TATGGAGTCCATGCACGAGA | Genscript |
| Kmt2d g1 | TCCGAAACATGTAAATACCG | https://www.ncbi.nlm.nih.gov/pmc/articles/PMC4148324/ |
| Kmt2d g2 | CTGTGCCCCTAACTGTGTAG | https://www.ncbi.nlm.nih.gov/pmc/articles/PMC4148324/ |
| Kmt2c g1 | TGCCAACCAGCACGCTTTAG | https://www.ncbi.nlm.nih.gov/pmc/articles/PMC4206212/ |
| Chd7 g1 | CCAGGGATGATGAGTCTTTT | https://www.ncbi.nlm.nih.gov/pmc/articles/PMC4148324/ |
| Ctdnep1 g1 | ACCCAAGCAAACTCACCGTG | http://chopchop.cbu.uib.no/ Labun et al., NAR 2016 |
| Ctdnep1 g2 | TTTGCGGAGGCAGATCCGCA | http://chopchop.cbu.uib.no/ Labun et al., NAR 2016 |
| Kbtbd4 g1 | TCTGAGCTGACAGGACCAGC | http://chopchop.cbu.uib.no/ Labun et al., NAR 2016 |
| Kbtbd4 g2 | CTTGGCAGCAGTGTAGAGCT | http://chopchop.cbu.uib.no/ Labun et al., NAR 2016 |
| Lrp1b g1 | CATTTGTCAAAACTGTGCAA | http://chopchop.cbu.uib.no/ Labun et al., NAR 2016 |
| Lrp1b g2 | TAATCCCGAGAGAGTAAGGA | http://chopchop.cbu.uib.no/ Labun et al., NAR 2016 |
| Ddx3x g1 | TACAGCAGTTTTGGATCACG | http://chopchop.cbu.uib.no/ Labun et al., NAR 2016 |
| Ddx3x g2 | CATACAGCAGTTTTGGATCA | http://chopchop.cbu.uib.no/ Labun et al., NAR 2016 |

**In vivo transfection**. For in vivo transfection, plasmid DNA and in vivo-jetPEI transfection reagent (Polyplus-transfection) were mixed according to the manufacturer's instructions. pPBase and piggyBac donor plasmids were mixed at a 1:4

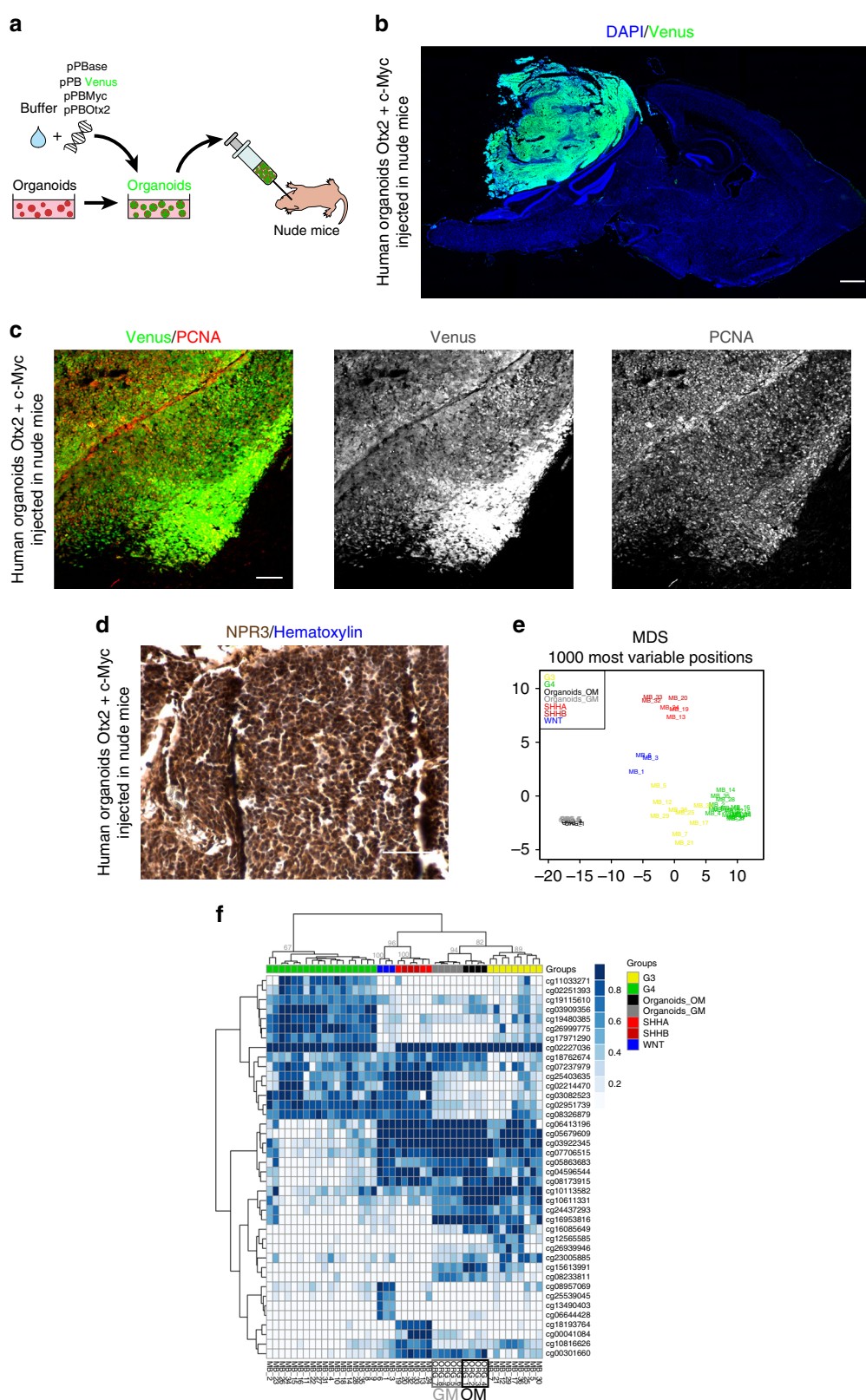

ratio. Plasmids encoding for sgRNAs targeting the same gene were mixed in equal amounts. The pPB CAG Venus plasmid was always co-transfected as a reporter. P0-P1 CD1 mice were anesthetized on ice for 2 min, placed on a stage in a stereotactic apparatus and medially injected at lambda: −3.6 D/V: −1.6 with 4 μl of transfection mix using a pulled glass capillary and a FemtoJet microinjector (Eppendorf). Animals were killed at 10 days, 3 months or at a humane endpoint as they displayed signs of morbidity. All experiments were done with all relevant ethical regulations for animal testing and research. The experiments were approved by the Italian Ministry of Health as conforming to the relevant regulatory standards.

**Immunofluorescence and immunohistochemistry.** Mice were intraventricularly perfused with 4% PFA, brains were dissected and post-fixed overnight in 4%

**Fig. 5 Cerebellar organoids electroporation with *Otx2/c-MYC* induces Group 3 MB in vivo. a** Schematic representation of in vivo injection of modified cerebellar organoids. **b** DAPI staining and GFP immunofluorescence (Venus) of the sagittal brain section of nude mouse 1 month after injection of human cerebellar organoids electroporated with pPBase + pPBMyc + pPBOtx2 + pPBVenus. **c** Confocal images of GFP (Venus) and PCNA immunofluorescence of tumors in nude mouse 1 month after injection of human cerebellar organoids electroporated with pPBase + pPBMyc + pPBOtx2 + pPBVenus. **d** NPR3 immunohistochemistry and Hematoxylin staining of tumor in nude mouse 1 month after injection of human cerebellar organoids electroporated with pPBase + pPBMyc + pPBOtx2 + pPBVenus. **e** MDS (multidimensional scaling) analysis performed on the 1000 most variable probes of the whole-genome DNA methylation data shows a close similarity between organoids and group 3 MBs. Color legend of the MDS plot as follows: OM Organoids (Organoids_OM, black); GM Organoids (Organoids_GM, gray); WNT, Wingless MB (blue); SHH-A, Sonic Hedgehog MB-adulthood and childhood (red); SHH-B Sonic Hedgehog MB infant (dark red); G3, Group 3 MB (yellow); G4, Group 4 MB (green). **f** Hierarchical clustering and heatmap of beta values relative to the 39 high-quality CpG islands better discriminating MB subgroup in the Hovestadt set (Hovestadt et al.)[35]. The heatmap shows normalized methylation levels in organoid samples and MB samples. Clusters were obtained by means of Ward's minimum variance method, using the Euclidean distance. Color legend: OM Organoids (Organoids_OM, black); GM Organoids (Organoids_ GM, gray); WNT, Wingless MB (blue); SHH-A, Sonic Hedgehog MB-adulthood and childhood (red); SHH-B Sonic Hedgehog MB infant (dark red); G3, Group 3 MB (yellow); G4, Group 4 MB (green). Scale bars 1 mm in (**b**), 100 μm in (**c**, **d**).

PFA. Brains were either cryoprotected in 30% (w/v) sucrose in water and embedded in Frozen Section Compound (Leica, 3801480), or embedded in paraffin (brains were dehydrated with ethanol, then kept sequentially in xylene and paraffin solutions). Spine tissues were decalcified for 3 weeks in 10% EDTA before being processed for embedding in Frozen Section Compound. Frozen Section Compound embedded brains were cryosectioned at 20–40 μm with a Leica CM 1850 UV Cryostat. Paraffin-embedded brains were sectioned using a Leica Microtome at 10 μm. Immunofluorescence stainings were performed on glass slides. Blocking and antibody solutions consisted of PBS supplemented with 3% goat serum, 0.3% Triton X-100 (Sigma). Primary antibodies were incubated overnight at 4 °C, and secondary antibodies for 1 h at room temperature. Nuclei were stained with 1 μg/ml DAPI (Sigma) or 1 μM DRAQ5 (ThermoFisher). Sections and coverslips were mounted with permanent mounting medium.

Immunohistochemistry stainings were performed on rehydrated paraffin sections. Antigen retrieval was performed by incubating slices for 30 min in retrieval solution (10 mM sodium citrate, 0.5% Tween-20 (v/v), pH 6.0) at 98 °C. Primary antibodies were incubated overnight at 4 °C, and secondary antibodies for 1 h at room temperature in antibody solution. ABC solution was used 2 h at room temperature (Vectastain Elite ABC Kit Standard PK-6100). The sections were incubated with the substrate at room temperature until suitable staining was observed (DAB Peroxidase Substrate Kit, SK-4100). Nuclei were counterstained with hematoxylin.

The used antibodies are listed below:

**Imaging**. Images were acquired with a Zeiss Axio Imager M2 (Axiocam MRc, Axiocam MRm), and for confocal imaging with either Leica TCS Sp5 or X-Light V2 confocal Imager optical. Images were processed using ImageJ software. Figures were prepared using Adobe Photoshop (Adobe).

**Genes identification and Smarca4 expression levels in G3 MB**. Genes that show differential expression (higher than 16-fold) compared with normal cerebellum have been identify using the online tool:

https://hgserver1.amc.nl/cgi-bin/r2/main.cgi?
&dscope&=&MB500&option&=&about_dscope#

Smarca4 expression levels in human G3 MB tumors were obtained from the Pfister-223-MAS5.0-u133p2 data set. Smarca4 expression levels in wild-type hES-derived neural stem cells were obtained from the data sets GSE9921 and GSE7178. Smarca4 expression levels in human normal adult cerebellum were obtained from the data set GSE3526. All data were normalized to the average Smarca4 expression level in either the GSE9921 or the GSE3526 data sets. Smarca4 mutational status in each G3 MB patient was identified using the somatic variants data collected in the same online tool.

**Genomic DNA extraction**. Organoids were lysed in lysis buffer (20 mM EDTA, 10 mM Tris, 200 mM NaCl, 0.2% Triton X-100, 100 μg/ml Proteinase K, pH 8.0) for 1 h at 37 °C. Genomic DNA was extracted with phenol–chloroform and precipitated with isopropanol.

| Primary antibodies | Host species | Dilution | Company | Reference |
|---|---|---|---|---|
| BARHL1 | Rabbit | 1:1000 (organoids) 1:500 (tissues) | Atlas Antibodies | HPA004809 |
| β3-tubulin | Mouse | 1:2000 | Thermofisher Scientific | MA1-118 |
| Calbindin-D-28K | Rabbit | 1:500 | Sigma Aldrich | C9848 |
| Cleaved Caspase-3 (D175) | Rabbit | 1:200 | Cell Signaling Technology | 9661 |
| c-MYC (Y69) | Rabbit | 1:200 | Abcam | ab32072 |
| FLAG M2 | Mouse | 1:100 | Sigma Aldrich | F1804 |
| GFAP | Rabbit | 1:200 | Sigma Aldrich | G9269 |
| GFP | Chicken | 1:2000 | Abcam | ab13970 |
| GFP | Rabbit | 1:500 | Thermofisher Scientific | A11122 |
| Ki67 | Rabbit | 1:500 | Abcam | ab15580 |
| NeuN | Rabbit | 1:2000 | EMD Millipore Corporation | ABN78 |
| NPR3 | Rabbit | 1:100 | Abcam | ab37617 |
| OLIG2 | Rabbit | 1:200 | Abcam | ab33427 |
| OTX2 | Mouse | 1:100 | Santa Cruz Biotechnology | sc-514195 |
| PCNA | Mouse | 1:2000 (organoids) 1:500 (tissues) | EMD Millipore Corporation | MAB424 |
| PH3 | Rat | 1:500 | Abcam | ab10543 |
| SMARCA4/BRG1 (G-7) | Mouse | 1:100 | Santa Cruz Biotechnology | sc-17796 |
| SKOR2 | Rabbit | 1:1000 | Atlas Antibodies | HPA046206 |
| SOX2 | Rabbit | 1:1000 (organoids) 1:500 (tissues) | Abcam | ab97959 |
| SOX9 | Rabbit | 1:4000 (organoids) 1:2000 (tissues) | EMD Millipore Corporation | AB5535 |

| Secondary antibodies | Dilution | Company | Reference |
|---|---|---|---|
| Alexa Fluor 488 goat anti-chicken IgY | 1:500 | Thermofisher Scientific | A11039 |
| Alexa Fluor 488 goat anti-rabbit IgG | 1:500 | Thermofisher Scientific | A11008 |
| Alexa Fluor 546 goat anti-mouse IgG | 1:500 | Thermofisher Scientific | A11030 |
| Alexa Fluor 647 goat anti-mouse IgG | 1:500 | Thermofisher Scientific | A21235 |
| Alexa Fluor 546 goat anti-rabbit IgG | 1:500 | Thermofisher Scientific | A11035 A11010 |
| Alexa Fluor 647 goat anti-rabbit IgG | 1:500 | Thermofisher Scientific | A21245 |
| Alexa Fluor 647 goat anti-rat IgG | 1:500 | Thermofisher Scientific | A21247 |
| Goat anti-Rabbit IgG-heavy and light chain Biotinylated | 1:250 | Bethyl Laboratories Inc. | A120-101B |
| Goat anti-Mouse IgG-heavy and light chain Biotinylated | 1:250 | Bethyl Laboratories Inc. | A90-116B |

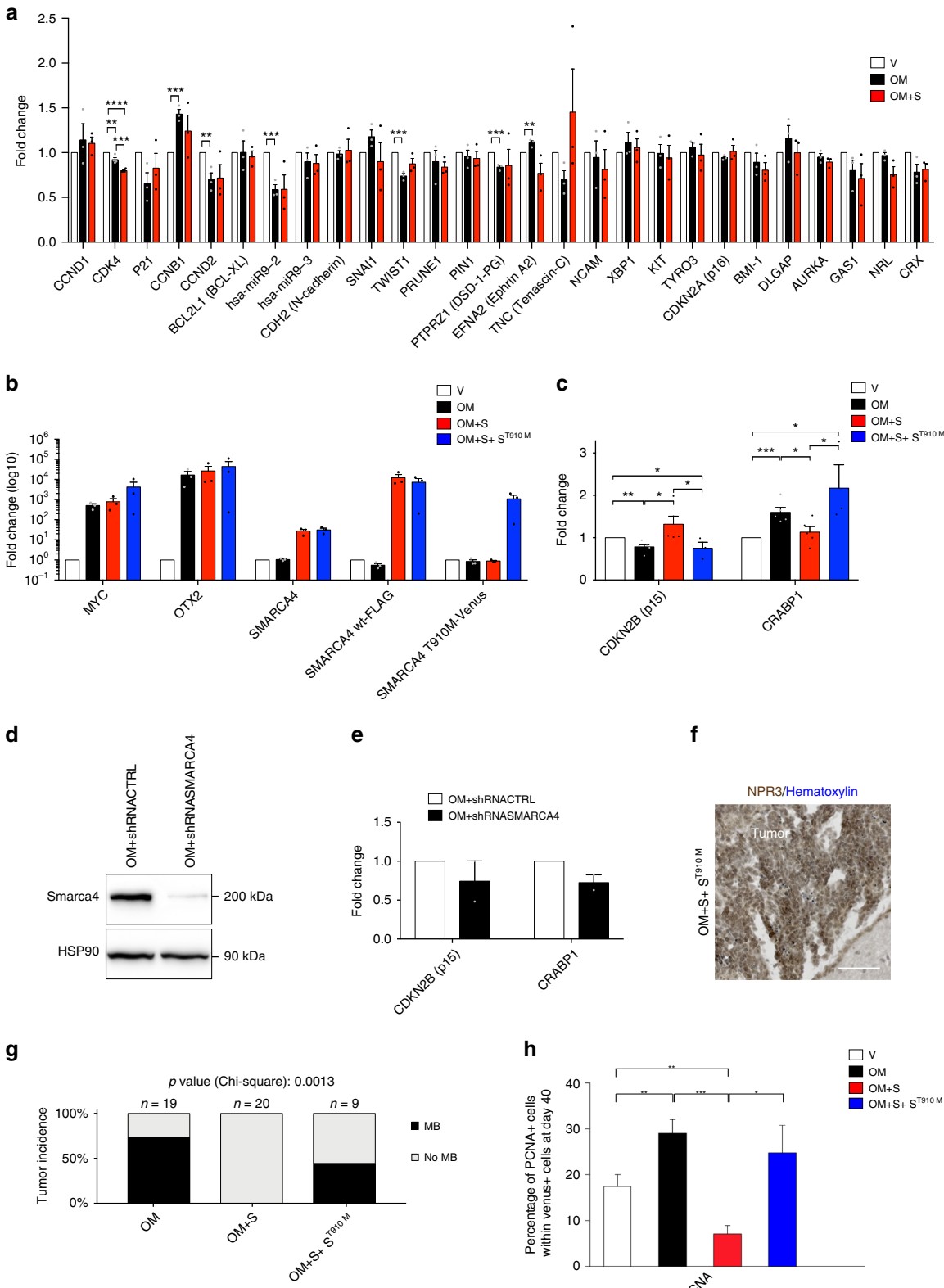

**In vivo bioluminescence imaging**. Mice were intraperitoneally administered 150 mg/kg D-Luciferin (Santa Cruz Biotechnology) 10 min before imaging. Animals were anesthetized with 2% isoflurane, and bioluminescent signal was captured using the In-vivo Xtreme system (Bruker). Mice were imaged weekly starting from 6 weeks after the transfection.

**Organoids maintenance, modification, and injection**. Human iPS cells (iPSC, ATCC-DYS0100) were maintained in self renewal on a layer of geltrex (Gibco,

A14133-01), in E8 Basal Medium (Gibco, A15169-01) supplemented with E8 supplement (50 ×). All cells were mycoplasma free. iPSC were dissociated with EDTA (Invitrogen) 0.5 mM, pH 8.0, for 3 min incubation, to maintain cell clusters. Cerebellar organoids were cultured as described by Muguruma et al.[20] Ishida et al.[19], and were electroporated at 35 days of differentiation protocol with 16.6 µg pCAG PiggyBac (PBase), 83.4 µg of pPB-YFP (Venus) and 16.6 µg pCAG PiggyBac (PBase), 16.6 µg of pPB-YFP, and either pPB CAG c-Myc (33.2 µg) + pPB CAG Gfi1 (33.2 µg)(GM) or pPB CAG c-Myc (33.2 µg) + pPB CAG Otx2 (33.2 µg)(OM) resuspended in Buffer 5 (under patent). Regarding Smarca4 experiment, organoids

**Fig. 6 Mutant SMARCA4 represses wild-type SMARCA4 functions. a** mRNA expression analysis of human cerebellar progenitors (AF22 cells) 72 h after nucleofecion with pPBase + pPBVenus (V), pPBase + pPBMyc + pPBOtx2 + pPBVenus (OM), pPBase + pPBMyc + pPBOtx2 + pPBSmarca4wt + pPBVenus (OM + S). **b, c** qRT-PCR analysis of human cerebellar progenitors (AF22 cells) 72 h after nucleofecion with pPBase + pPBVenus (V), pPBase + pPBMyc + pPBOtx2 + pPBVenus (OM), pPBase + pPBMyc + pPBOtx2 + pPBSmarca4wt + pPBVenus (OM + S), pPBase + pPBMyc + pPBOtx2 + pPBSmarca4wt + pPBSmarca4 T910M + pPBVenus (OM + S + ST910M). **d** Western blot analysis of human cerebellar progenitors (AF22 cells) 72 h after nucleofecion with pPBase + pPBMyc + pPBOtx2 + pPBVenus + pSCV2shCTRL (OM + shRNACTRL), pPBase + pPBMyc + pPBOtx2 + pSCV2shSMARCA4 (OM + shRNASMARCA4). **e** qRT-PCR analysis of human cerebellar progenitors (AF22 cells) electroporated with pPBase + pPBMyc + pPBOtx2 + pPBVenus + pSCV2shCTRL (OM + shRNACTRL), pPBase + pPBMyc + pPBOtx2 + pSCV2shSMARCA4(OM + shRNASMARCA4). **f** NPR3 immunohistochemistry and hematoxylin staining of tumors in CD1 mice after transfection with pPBase + pPBMyc + pPBOtx2 + pPBSmarca4wt + pPBSmarca4 T910M + pPBVenus (OM + S + ST910M). **g** Histograms show the percentage of mice that develop MB (3 months) after transfection at P0 with either OM, OM + S or OM + S + ST910M. **h** Quantification of cerebellar organoids GFP + /PCNA + cells at day 40 electroporated at day 35 with PBase + pPBVenus (V), pPBase + pPBMyc + pPBOtx2 + pPBVenus (OM), pPBase + pPBMyc + pPBOtx2 + pPBSmarca4wt + pPBVenus (OM + S), pPBase + pPBMyc + pPBOtx2 + pPBSmarca4wt + pPBSmarca4 T910M + pPBVenus (OM + S + ST910M). **a, b, c** At least $n = 3$ biologically independent experiments. **e** $n = 2$ biologically independent experiments. **h** $n = 6-11$ biologically independent organoids. Error bars in (**a, b, c, e, h**) represent standard error of the mean. Paired Student's $t$ test, one tail (**a, b, c, e**), two tails (**h**). *$p$-value < 0.05, **$p$-value < 0.01. ***$p$-value < 0.001. ****$p$-value < 0.0001. Chi-square test (**g**). Scale bar 100 µm in (**f**).

were electroporated with 16.6 µg pCAG PiggyBac (PBase), 16.6 µg of pPB-YFP, PB CAG c-Myc (22.2 µg), pPB CAG Gfi1 (22.2 µg) (GM), and pPB CAG Smarca4-IRES-GFP ((22.2 µg) WT or mutant) or pPB CAG c-Myc (22.2 µg), pPB CAG Otx2 (22.2 µg) (OM), and pPB CAG Smarca4-IRES-GFP ((22.2 µg) WT or mutant), or GM/OM with pPB CAG Smarca4-IRES-GFP WT and mutant (11.1 µg). Organoids were transferred inside the Electroporation cuvettes (VWR, ECN 732-1136, 2 mm), and electroporation was performed with the Gene Pulser XcellTM. Twenty-four to 120 days after electroporation, OM and GM organoids were dissociated into Neurobasal Medium (Gibco, cat.21103049) and injected intracranially into nude mice (P04-P08) with a 30-gauge Hamilton Syringe. The mice were killed at the experimental endpoint (loss of weight, ataxia phenotype, suffering phenotype, kyphoscoliosis) and intraventricularly perfused with 4% PFA, and brains were collected, cryoprotected in 30% sucrose and embedded in Frozen Section Compound (Leica, 3801480). Brains were cryosectioned at 20–40 µm with Leica CM 1850 UV Cryostat.

About the drug screening, organoids were treated from days 37 to 57 of differentiation, with two doses of 5 µM 3-deazaneplanocin A (DZNep, Selleckchem) or DMSO. Organoids were treated from days 36 to 41 of differentiation, with one dose of either 5 µM Tazemetostat (EPZ-6438) or 5 µM GSK-126 or DMSO. After the drug treatment, organoids were fixed with in PFA 4%, cryoprotected in 20% sucrose and embedded in Frozen Section Compound (Leica, 3801480). Organoids were cryosectioned at 40 µm with Leica CM 1850 UV Cryostat.

**Histopathological evaluation**. Two $Otx2 + c\text{-}MYC$ mouse tumors were diagnosed by neuropathologists Francesca Gianno and Felice Giangaspero. In addition to standard hematoxylin and eosin staining, immunostaining was done on formalin-fixed paraffin-embedded tumors after dewaxing and rehydrating slides.

**AF22 culture and nucleofection**. Human iPSC-derived neuroepithelial-like stem cells AF22 were cultured in a 1:1 ratio mixture of neurobasal and DMEM/F12 media supplemented with N2 (1:100), B27 (1 µl/ml), 10 ng/ml EGF and 10 ng/ml FGF2. In total, $2 \times 10^6$ AF22 cells were nucleofected with 20 µg of plasmid DNA in 200 µl of nucleofection buffer using the T-020 program and a Nucleofector 2b device (Amaxa). Cells were collected after 72 h for further analyses. All cells were mycoplasma free.

**RNA isolation and qRT-PCR**. The total RNA was extracted with TRIzol Reagent (Invitrogen) and reverse transcribed using iScript cDNA synthesis kit (Biorad) according to the manufacturer's instructions. Quantitative PCR was performed using Power SYBR Green PCR Master Mix (Applied Biosystems). The results are presented as linearized Ct values normalized to the housekeeping gene GAPDH and the indicated reference value (2-ΔΔCt). Primer sequences used for qRT-PCR are listed:

| Target Gene | Forward primer | Reverse primer |
|---|---|---|
| CCND1 | ATGTTCGTGGCCTCTAAGATGA | CAGGTTCCACTTGAGCTTGTTC |
| CDK4 | CTTTGGCAGCTGGTCACATGG | CTCAGATCAAGGGAGACCCTCAC |
| P21 | CTGGAGACTCTCAGGGTCGAA | GATTAGGGCTTCCTCTTGGAGAA |
| CDKN2B (p15) | CCCTCGACACTCACCATGAA | CGACCCCTGGAATGTCACAC |
| GAPDH | CCACTCCTCCACCTTTGAC | ACCCTGTTGCTGTAGCCA |
| CCNB1 | ATGTGCCCCTGCAGAAGAAG | TTTCCAGTGACTTCCCGACC |
| GAS1 | GGACGAGAACTGCAAGTCCA | AGACTTTGCCGCAGTAGGTC |
| hsa-miR9-2 | CTAACGCTGCCGGAGATTAC | TACTTGCCGCGCTTAAGATT |
| hsa-miR9-3 | GCGCTCGAGGCTCTCTAAG | GAGGGGATGGACAGACACAC |
| CDH2 (N-cadherin) | CGACGAATGGATGAAAGACCC | GCCACTGCCTTCATAGTCAAAC |
| SNAI1 | CTAGGCCCTGGCTGCTACAA | CCTGGCACTGGTACTTCTTGA |
| TWIST1 | CTTCTCGGTCTGGAGGATGG | GAAACAATGACATCTAGGTCTCCG |
| PRUNE1 | CACTGAGCAGATGCTGAGAAAAG | TGCACTAATGGCCACCTTGAC |
| PIN1 | GAGAAGATCACCCGGACCAA | AAAGTCCTCCTCTCCCGACT |
| NRL | CATTGGGGCTGAGTCCTGAA | CGCACAGACATCGAGACCA |
| CRX | CCCACTATTCTGTCAACGCCT | GACGTCTGGGTACTGGGTCT |
| CCND2 | GCGGAGAAGCTGTGCATTTA | CATGCTTGCGGATCAGAGAC |
| BCL2L1 (BCL-XL) | CTGTGCGTGGAAAGCGTAGA | GCTGCTGCATTGTTCCCATAG |
| NCAM | GAGATCAGCGTTGGAGAGTCC | GGAGAACCAGGAGAGTGTCTTTATCTT |
| PTPRZ1 (DSD-1-PG) | TGCAGAGCTGTACTGTTGACTT | CTGTGCTAGCTTAACCCTGCT |
| EFNA2 (Ephrin A2) | CTACATCTCTGCCACGCCTC | CGGGCTGCTACACGAGTTAT |
| TNC (Tenascin-C) | TCTGGTGCTGAACGAACTGC | CCAGGAAACTGTGAACCCGTA |
| XBP1 | TGCCAGAGATCGAAAGAAGGC | CCAAGCGCTGTCTTAACTCCT |
| KIT | CCTGAACACCAGCAGTGGAT | TGTAAGTGCCTCCTTCGGTG |
| TYRO3 | GCCACTGGTGGTCTCTTCTC | CGTTAGCACACCAAGGACCA |
| CDKN2A (p16) | CTTCGGCTGACTGGCTGG | CGTGTCCAGGAAGCCCTC |
| CRABP1 | ACTTCAAGGTCGGAGAAGGC | AGTTTGCGTGCAGTGGATCT |
| BMI-1 | CGCTTGGCTCGCATTCATTT | CACACACATCAGGTGGGGAT |
| DLGAP5 (HURP) | TGTTTGGTTGAGGTTTCACGC | CCTGTGTCGACTGGCAAAATG |
| AURKA | CAGTACATGCTCCATCTTCCAG | AAAGAACTCCAAGGCTCCAG |
| cMyc | GCGACTCTGAGGAGGAACAA | CCTCCAGCAGAAGGTGATCC |
| OTX2 | GAGGTGGCACTGAAAATCAAC | TCTTCTTTTTGGCAGGTCTCA |
| SMARCA4 | TGCTCCGACGACTCAAGAAG | TTCATCAGGGTCTTGGTGCC |
| SMARCA4-FLAG | GAAGGAGAAGGCACAGGACC | TCATCGTCGTCCTTGTAGTCG |
| SMARCA4-Venus | GAAGGAGAAGGCACAGGACC | CACCACCCCGGTGAACAG |

**Western blot**. Proteins were extracted from AF22 cells and fresh-frozen tumor tissues in lysis buffer (50 mM Tris-HCl, 150 mM NaCl, 2 mM $MgCl_2$, 0.5% NP-40, 50 mM NaF, 1 mM $Na_3VO_4$, 1 mM PMSF, supplemented with proteases inhibitors and 0.2 mg/ml DNase, pH 7.4). Proteins were separated by SDS-PAGE and transferred onto a PVDF membrane. The membrane was blocked in 5% low-fat milk/TBST for 1 h at room temperature, and probed with primary antibodies overnight ad 4 °C. Secondary antibodies were incubated for 1 h at room temperature. Protein levels were detected using the Clarity Western ECL Substrate (Biorad). Antibodies used are listed here:

| Primary antibodies | Host species | Dilution | Company | Reference |
|---|---|---|---|---|
| ARID1A/BAF250A | Rabbit | 1:1000 | Cell Signaling Technology | 12354 |
| HSP90 | Mouse | 1:5000 | Santa Cruz Biotechnology | sc-13119 |
| SMARCA2/BRM | Rabbit | 1:1000 | Cell Signaling Technology | 11966 |
| SMARCA4/BRG1 (G-7) | Mouse | 1:1000 | Santa Cruz Biotechnology | sc-17796 |
| SMARCB1/BAF47 | Rabbit | 1:1000 | Cell Signaling Technology | 91735 |
| SMARCC1/BAF155 | Rabbit | 1:1000 | Cell Signaling Technology | 11956 |
| SMARCC2/BAF170 | Rabbit | 1:1000 | Cell Signaling Technology | 12760 |
| Vinculin | Mouse | 1:1000 | Santa Cruz Biotechnology | sc-73614 |
| **Secondary antibodies** | | **Dilution** | **Company** | **Reference** |
| Peroxidase AffiniPure Goat Anti-Rabbit IgG (H + L) | | 1:5000 | Jackson ImmunoResearch | 111-035-003 |
| Peroxidase AffiniPure Goat Anti-Mouse IgG (H + L) | | 1:5000 | Jackson ImmunoResearch | 115-035-003 |

**Tumor spheroids, drug treatment, and flow cytometry analysis**. Tumors were dissected aseptically from two CD1 mice injected at P0 with pPBase + pPBc-Myc + pPBOtx2 + pPBVenus when they displayed signs of morbidity. Tumor cells were disaggregated to a single-cell suspension using diluted Accutase (1:3) and cultured in a 1 : 1 ratio mixture of neurobasal and DMEM/F12 media supplemented with N2 (1:100), B27 (1:50), 20 ng/ml EGF and 20 ng/ml FGF2. Cells were grown in tumor spheroids in ultralow adherent plates and passaged weekly. For the drug treatment, tumor spheroids were dissociated to single cells using diluted Accutase (1:3), and 5000 cells/ml were seeded in the culture medium. Cells were allowed to grow into tumor spheroids for 3 days. Subsequently, spheroids were treated for further 1–3 days with 0.5, 1, 2.5, 5 µM 3-deazaneplanocin A (DZNep, Selleckchem), 10 nM Panobinostat (LBH589, Selleckchem), 5 µM Tazemetostat (EPZ-6438, Selleckchem), 5 µM GSK-126 (Selleckchem), or DMSO. After the drug treatment, brightfield and fluorescence images of the tumor spheroids were acquired, and tumor spheroid area was quantified using ImageJ. Spheroids were eventually collected, fixed in 4% PFA for 3 h at 4 °C and stained as the immunostaining protocol done also for mouse tissues.

Tumor spheroids for flow cytometry were harvested after drug treatment, and dissociated with 0.25% Trypsin-EDTA (Gibco) to a single-cell suspension. Cells for analysis of apoptosis were incubated for 30 min in culture medium to allow recovery from dissociation, then resuspended in PBS and stained with LIVE/DEAD Fixable Near-IR Stain (Invitrogen) for 20 min. Subsequently, cells were resuspended in annexin-binding buffer (10 mM HEPES, 140 mM NaCl, 2.5 mM $CaCl_2$, pH 7.4) and stained for 15 min with Annexin V, Alexa Fluor 647 conjugate (Invitrogen). Cells for DNA content analysis were fixed with ice-cold 70% v/v ethanol at −20 °C for at least 20 min. Fixed cells were stained with 10 µg/ml propidium iodide (Invitrogen) and treated with 100 µg/ml PureLink RNase A (Invitrogen) for 30 min at 37 °C. Cells were analyzed using a FACSCanto (BD Biosciences) and data analysis was performed using FlowJo software (LLC).

**Survival analysis**. Survival analysis was performed calculating the lifespan in days of every mice injected with each specific combination of plasmids. Mice which were killed at the experimental endpoint or died due to undetermined causes during the study were censored in the analysis. Data were displayed using the Kaplan–Meier format, and statistical significance of the results was tested using the Log-rank (Mantel–Cox) test.

**Statistical analysis**. Organoids quantitative analysis: the data are presented as mean + s.e.m., for each condition 5–11 organoids were examined and at least 110–200 cells were quantified. Organoids electroporated with pPBase + pPBVenus (encoding for Venus) were considered as the positive control, and were used to set the parameters for cells count on ImageJ software. Data were compared using a unpaired Student's $t$ test, two tails.

Tumor Spheroids data are presented as mean + s.d. of two biologically independent ex vivo experiments. In total, 120 spheroids were counted for each condition. After the drug treatment, brightfield and fluorescence images of the tumor spheroids were acquired, and tumor spheroid area was quantified using ImageJ. Data were compared using a unpaired Student's $t$ test, two tails with Welch's correction.

**DNA methylation profiling**. DNA methylation profiling was performed, according to protocols approved by the institutional review board with written consent obtained from the patients' parents. Tumor areas with highest tumor cell content ( ≥ 70%) were selected for DNA extraction. Samples were analyzed using Illumina Infinium HumanMethylationEPIC BeadChip (EPIC) arrays according to the manufacturer's instructions, on Illumina iScan Platform. In detail, 250 ng or 500 ng of DNA was used as input material for fresh-frozen or formalin-fixed paraffin-embedded tissues, respectively. Generated methylation data were compared with the Heidelberg brain tumor classifier[34] (http://molecularneuropathology.org) to assign a subgroup score for the tumor compared with 91 different brain tumor entities. All tumors had a score of at least 0.8 in the reported methylation class.

EPIC BeadChip data were analyzed by means of R (V .3.4.3), using different packages: ChAMP pipeline (V.2.9.9)[71] for quality checks and filters, to calculate methylation levels and functionally annotate probes at the gene level. Multidimensional scaling (MDS) on the cohort samples was performed using cmdscale function, with Euclidean distance. Heatmap depicting normalized beta values was created by means of pheatmap function, using Ward's minimum variance method[72], and Euclidean distance to cluster samples and probes. Low-quality CpG islands among the 48 ones identified from Hovestadt et al.[35] were removed from the analysis. Finally, bootstrap analyses were carried out using pvclust package[73].

**Gene expression analysis of electroporated organoids**. Two biological replicates for each group (Untreated or Electroporated) were used. The total organoids RNAs were extracted using Trizol reagent (Invitrogen), subjected to DNase-I (Ambion) treatment and RNAs were depleted of ribosomal RNA. Sequencing libraries for whole transcriptome analysis were prepared using Stranded mRNA-Seq Library Preparation Kit. RNA-seq was performed on an Illumina HiSeq 2500 Sequencer using standard conditions at the Next-Generation Sequence Facility of University of Trento (CIBIO). The obtained reads were 75 bp long, paired ends, and 30 Millions on average for each sample. Quality control analysis was performed using FastQC (www.bioinformatics.babraham.ac.uk/projects/fastqc/). All the sample sequence reads were mapped with STAR aligner (v2.5.3) using recommend parameters. To provide an estimate of gene expression and compute differential gene expression, the reads were proportionally assigned to the human gene transcripts (ENSEMBL HG38), based on the mappings using HT-SEQ count (http://www-huber.embl.de/users/anders/HTSeq. Differential gene expression analysis was performed using the gene raw counts, within the R/Bioconductor edgeR package. The differential gene expression pipeline within the edgeR package was customized to estimate the dispersion parameter for each library using the biological group dispersion and identify DE genes between treated versus the control samples. log2(fold change) ≥ 1 and baseMean > 3 CPM were considered for differentially regulated genes and the $p$-value was adjusted for multiple testing using the Benjamini–Hochberg correction with a false discovery rate (FDR) ≤ 0.05. Differentially expressed gene lists obtained from low-level procedures were analyzed for functional associations. Data were analyzed through DAVID Bioinformatics Resources v6.8 using the suggested standard parameters.

**Reporting summary**. Further information on research design is available in the Nature Research Reporting Summary linked to this article.

## Data availability
DNA Methylation Raw Data and Classifier Results. The data sets generated during and/or analyzed during the current study are available in GEO: GSE128218. Token: sxefoyqubnivpkh.

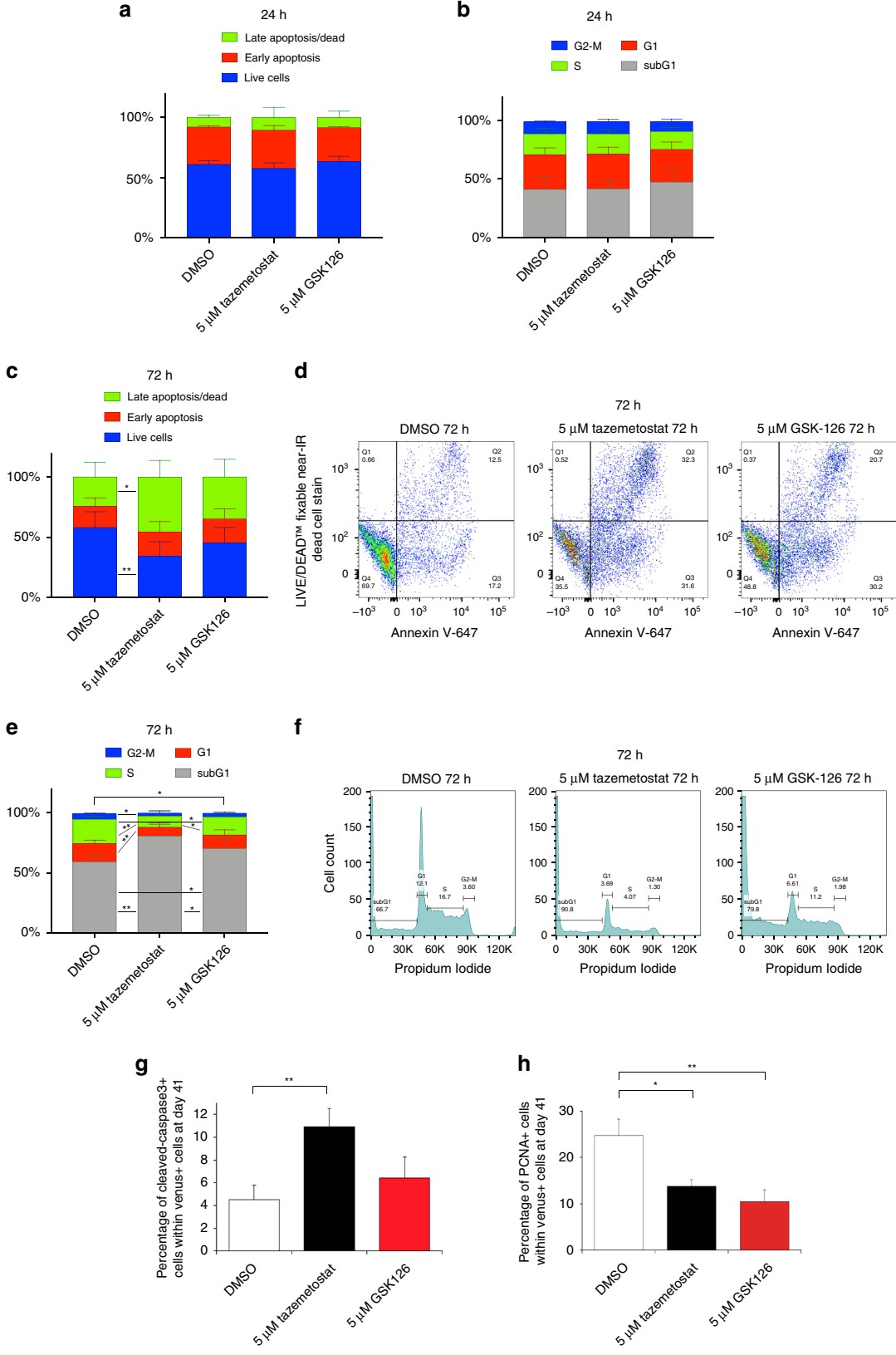

**Fig. 7 EZH2 inhibition with Tazemetostat increases cell death of OM-derived tumor spheroids and in MB organoids. a** Histograms show FACS analysis of OM-induced tumor spheroids cell death (late and early apoptosis) after 1 day of drug treatment (DMSO, Tazemetostat, GSK-126). **b** Histograms show FACS analysis of OM-induced tumor spheroids (cell cycle analysis) after 1 day of drug treatment (DMSO, Tazemetostat, GSK-126). **c** Histograms show FACS analysis of OM-induced tumor spheroids cell death (late and early apoptosis) after 3 days of drug treatment (DMSO, Tazemetostat, GSK-126). **d** Representative FACS analysis of OM-induced tumor spheroids cell death (late and early apoptosis) after 3 days of drug treatment (DMSO, Tazemetostat, GSK-126). **e** Histograms show FACS analysis of OM-induced tumor spheroids (cell cycle analysis) after 3 days of drug treatment (DMSO, Tazemetostat, GSK-126). **f** Representative FACS analysis of OM-induced tumor spheroids (cell cycle analysis) after 3 days of drug treatment (DMSO, Tazemetostat, GSK-126. **g** Quantification of cerebellar organoids at day 41, electroporated at day 35 with pPBase + pPBMyc + pPBOtx2 + pPBVenus (OM), and treated for 5 days with either DMSO or Tazemetostat or GSK-126. Percentage of active caspase-3-positive cells between Venus-positive cells. **h** Quantification of cerebellar organoids cells at day 41, electroporated at day 35 with pPBase + pPBMyc + pPBOtx2 + pPBVenus (OM), and treated for 5 days with either DMSO or Tazemetostat or GSK-126. Percentage of PCNA-positive cells between Venus-positive cells. **a** $n = 2$ biologically independent experiments. (**b**, **c**, **e**) $n = 5$ biologically independent experiments. **g**, **h** $n = 5$–11 biologically independent organoids. Error bars in (**a**, **b**, **c**, **e**) represent standard deviation, error bars in (**g**, **h**) represent standard error of the mean. Paired Student's $t$ test, two tails. *$p$-value < 0.05, **$p$-value < 0.01.

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

## Acknowledgements

We thank Prof. Alessandro Quattrone, Alessia Soldano, Alessio Zippo, Francesco Antonica, Alberto Inga, Luciano Conti, Pierre Vanderhaeghen, Bassem Hassan, Cedric Blanpain, Luca Fava, Andrea Lunardi and members of CIBIO for helpful discussions and advices. We thank Sergio Robbiati (MOF facility). We thank Prof. Anthony Imbalzano for pBS hBRG1 (Smarca4). This work was funded by grant from The Giovanni Armenise-Harvard foundation (Career Development Award to L.T.), My First AIRC Grant to L.T., CARITRO to L.T., Ministry of Health, Ricerca Corrente to E.M. and A.C.; AIRC IG 21614 to M.T.

## Authors contributions

C.B., M.G., G.A. and D.C., performed in vivo experiments. M.A., C.L. and M.C. performed organoids experiments. S.P. performed the electroporated organoids Bioinformatics analyses. C.B., M.A. and L.T. designed and analyzed all experiments and wrote the paper. E.M., E.F., A.C., L.P., A.M., M.T. and F.L. performed DNA methylation and analysis of OM electroporated organoids injected in nude mice. Fe.G. and Fr.G. performed immunohistochemical analysis of mouse MB. C.B. created Figs. 4a, 5a, and Supplementary Fig. 9A.

## Competing interests

The authors declare no competing interests.
