## [Peer Review File · Nature Communications]

Reviewers' Comments:

Reviewer #1:

Remarks to the Author:

In the revised version, the authors have made several changes to the text and added additional data to address the comments I made before. My concerns have been sufficiently addressed.

Reviewer #2:

Remarks to the Author:

The authors have done an excellent job at revising their manuscript to address my previous concerns. The manuscript still needs better editing with respect to the flow of the text (page numbers in the manuscript would also have helped reviewers). Currently there are nearly no paragraph breaks anywhere in the paper, except after a subheader. Each new topic/idea/thread should begin a new paragraph. For instance, the Introduction is one long paragraph - this needs to be broken up. Likewise in the results, for examples: page one - paragraph break, 5th line from bottom ("Several gene combinations..."), page two - paragraph break, 7 lines from bottom of the first paragraph ("Here, we have..."); page 3, 14 lines from bottom of first paragraph ("To better characterize...").

Reviewer #3:

Remarks to the Author:

The authors addressed the concerns from the original review sufficiently. The new results with Smarca4(T910M)+WT over expression experiments demonstrate that some of the missense mutations identified in the patient tumors may abrogate the function of the WT protein in MB formation, effectively removing some functions of the SWI/SNF complex to allow Otx2/c-myc tumorigenesis to occur. It also shows why mouse models where smarca4 is removed randomly with CRISPR may be insufficient to model these events in human cells, as specific point mutations are required for this effect.

Dear Dr. Lombardo,

We thank the reviewers for their careful reading of the manuscript and their constructive remarks.

We amended the manuscript in agreement with their suggestions.

REVIEWERS' COMMENTS:

Reviewer #1 (Remarks to the Author):

In the revised version, the authors have made several changes to the text and added additional data to address the comments I made before. My concerns have been sufficiently addressed.

We thank Reviewer#1 for his/her remarks.

Reviewer #3 (Remarks to the Author):

The authors have done an excellent job at revising their manuscript to address my previous concerns. The manuscript still needs better editing with respect to the flow of the text (page numbers in the manuscript would also have helped reviewers). Currently there are nearly no paragraph breaks anywhere in the paper, except after a subheader. Each new topic/idea/thread should begin a new paragraph. For instance, the Introduction is one long paragraph - this needs to be broken up. Likewise in the results, for examples: page one - paragraph break, 5th line from bottom ("Several gene combinations..."), page two - paragraph break, 7 lines from bottom of the first paragraph ("Here, we have..."); page 3, 14 lines from bottom of first paragraph ("To better characterize...").

We thank Reviewer#3 for his/her remarks. We have inserted breaks in the paper as suggested.

Reviewer #4 (Remarks to the Author):

The authors addressed the concerns from the original review sufficiently. The new results with Smarca4(T910M)+WT over expression experiments demonstrate that some of the missense mutations identified in the patient tumors may abrogate the function of the WT protein in MB formation, effectively removing some functions of the SWI/SNF complex to allow Otx2/c-myc tumorigenesis to occur. It also shows why mouse models where smarca4 is removed randomly with CRISPR may be insufficient to model these events in human cells, as specific point mutations are required for this effect.

We thank Reviewer#4 for his/her remarks.